# LinearSR: Unlocking Linear Attention for Stable and Efficient Image Super-Resolution

**Xiaohui Li**[1,2,♠]   **Shaobin Zhuang**[1]   **Shuo Cao**[3,2]   **Yang Yang**[4,2]
**Yuandong Pu**[1,2]   **Qi Qin**[2]   **Siqi Luo**[1,2]   **Bin Fu**[2]   **Yihao Liu**[2,*]

[1] Shanghai Jiao Tong University  [2] Shanghai Artificial Intelligence Laboratory
[3] University of Science and Technology of China  [4] The Australian National University

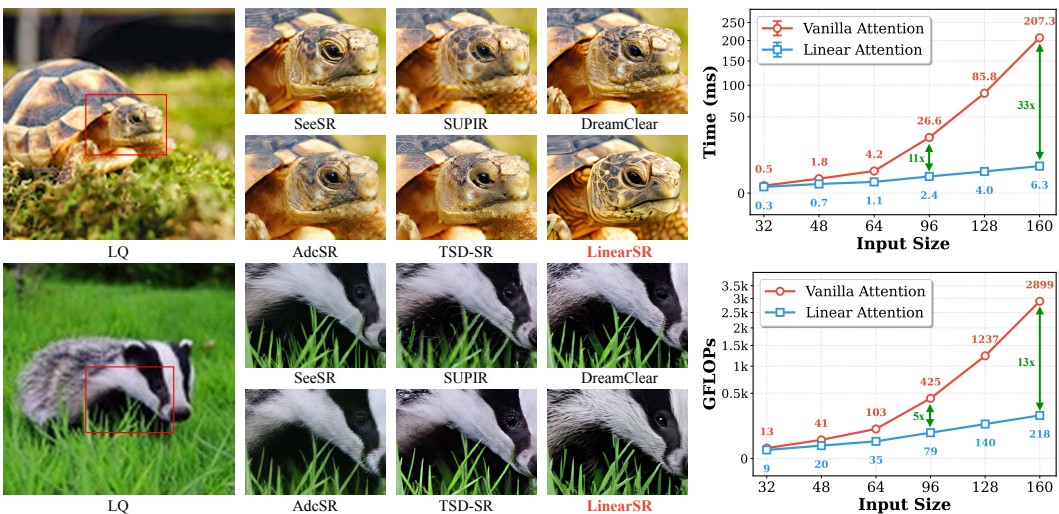

Figure 1: **LinearSR enables high-fidelity super-resolution at a linear computational cost.** Left: LinearSR produces high-fidelity visual results, restoring fine details and textures. Right: The plots highlight the dramatic efficiency advantage of our Linear Attention. As input size grows, its cost in time and GFLOPs scales linearly, versus the quadratic growth of vanilla attention.

## ABSTRACT

Generative models for Image Super-Resolution (SR) are increasingly powerful, yet their reliance on self-attention's quadratic complexity ($O(N^2)$) creates a major computational bottleneck. Linear Attention offers an $O(N)$ solution, but its promise for photorealistic SR has remained largely untapped, historically hindered by a cascade of interrelated and previously unsolved challenges. This paper introduces **LinearSR**, a holistic framework that, for the first time, systematically overcomes these critical hurdles. Specifically, we resolve a fundamental, training instability that causes catastrophic model divergence using our novel "knee point"-based Early-Stopping Guided Fine-tuning (ESGF) strategy. Furthermore, we mitigate the classic perception-distortion trade-off with a dedicated SNR-based Mixture of Experts (MoE) architecture. Finally, we establish an effective and lightweight guidance paradigm, TAG, derived from our "precision-over-volume" principle. Our resulting LinearSR model simultaneously delivers state-of-the-art perceptual quality with exceptional efficiency. Its core diffusion forward pass (**1-NFE**) achieves **SOTA**-level speed, while its overall multi-step inference time remains highly competitive. This work provides the first robust methodology for applying Linear Attention in the photorealistic SR domain, establishing a foundational paradigm for future research in efficient generative super-resolution.

---

*Corresponding author. ♠This work was done during internship at Shanghai AI Laboratory.

# 1 INTRODUCTION

Recent advancements in Image Super-Resolution (SR) are dominated by generative models (Chen et al., 2023; Rombach et al., 2022; Duan et al., 2025) that leverage the powerful self-attention mechanism to synthesize photorealistic details. However, this power comes at a steep price: the quadratic complexity ($O(N^2)$) of self-attention imposes a major computational bottleneck. Linear Attention (Shen et al., 2021; Katharopoulos et al., 2020; Wang et al., 2020; Cai et al., 2022), with its $O(N)$ complexity, has emerged as a compelling alternative. Its potential was successfully shown in general image generation by models like SANA (Xie et al., 2024), which validated its ability to capture global dependencies efficiently. This work addresses the central challenge: *how can the efficiency of Linear Attention be fully unlocked to satisfy the extreme fidelity requirements of super-resolution?*

Translating this theoretical promise into practice, however, required overcoming a significant cascade of technical hurdles. Our initial exploration into guidance was driven by the scarcity of high-resolution images paired with high-precision annotations. This motivated us to test information-agnostic extractors like DINO, whose surprising success led us to the "precision-over-volume" principle, which was ultimately validated by the concise TAG model. Subsequently, a more formidable hurdle soon emerged: a critical training instability. When fine-tuning a converged model–a standard industry practice–the loss would abruptly diverge to NaN, halting all progress and revealing a fundamental flaw in applying conventional methods to Linear Attention SR. Finally, even after resolving this, a persistent final barrier remained: the classic perception-distortion trade-off. The model struggled to improve perceptual realism (e.g., finer textures) without simultaneously sacrificing reconstruction fidelity (e.g., PSNR), making it the last obstacle to unlocking top-tier performance.

To conquer these challenges, we propose **LinearSR**, a framework designed to harmonize efficiency, stability, and performance. As encapsulated by our teaser in Fig. 1, LinearSR achieves two goals: it produces photorealistic results while demonstrating an efficiency advantage. The plot highlights our model's linear ($O(N)$) scaling, in stark contrast to the quadratic ($O(N^2)$) cost of standard attention.

Crucially, this linear scaling advantage is not merely theoretical but is directly reflected in the performance of the core architecture, independent of orthogonal optimizations like model distillation. Specifically, for megapixel-scale synthesis (1024×1024), our model's fundamental diffusion forward pass (1-NFE) sets a new **SOTA-level time of 0.036s**. This metric precisely benchmarks our structural contribution to the attention mechanism's efficiency. Consequently, the overall multi-step inference time remains highly competitive at 0.830s, demonstrating the practical viability of our approach. This achievement is built upon a triad of core contributions: (i) an Early-Stopping Guided Fine-tuning (ESGF) strategy that resolves the critical training instability; (ii) an SNR-based Mixture of Experts (MoE) architecture to mitigate the perception-distortion trade-off; and (iii) the adoption of the effective TAG-based guidance paradigm, validated by our "precision-over-volume" principle.

Equipped with this framework, LinearSR sets a new powerful benchmark for efficiency in the core diffusion forward pass. This work, for the first time, provides a robust and repeatable methodology to successfully apply Linear Attention in the high-fidelity SR domain. By establishing this foundational paradigm, we pave the way for numerous future optimizations, such as model distillation, to further push the boundaries of both speed and perceptual quality in generative super-resolution.

# 2 RELATED WORK

The paradigm in image restoration has shifted from traditional methods towards powerful diffusion-based generative priors (Lin et al., 2024; Wu et al., 2024b; Yu et al., 2024; Ai et al., 2024). While these models achieve state-of-the-art perceptual quality, their prohibitive computational cost spurred a new line of research focused on acceleration through methods like knowledge distillation and diffusion inversion (Wang et al., 2024b; Wu et al., 2024a; Dong et al., 2025; Yue et al., 2025; Chen et al., 2025a). However, these post-hoc optimizations do not resolve the fundamental architectural bottleneck: the quadratic complexity of the standard self-attention mechanism, which remains a severe bottleneck for high-resolution inputs. To address this limitation, linear attention methods offer a compelling $O(N)$ alternative. Pioneered in NLP (Wang et al., 2020; Katharopoulos et al., 2020) and successfully extended to other vision tasks (Shen et al., 2021; Cai et al., 2022) and generative modeling (Xie et al., 2024), this approach provides a strong foundation. Yet, translating this theoretical efficiency to the demanding super-resolution task has proven notoriously difficult, historically

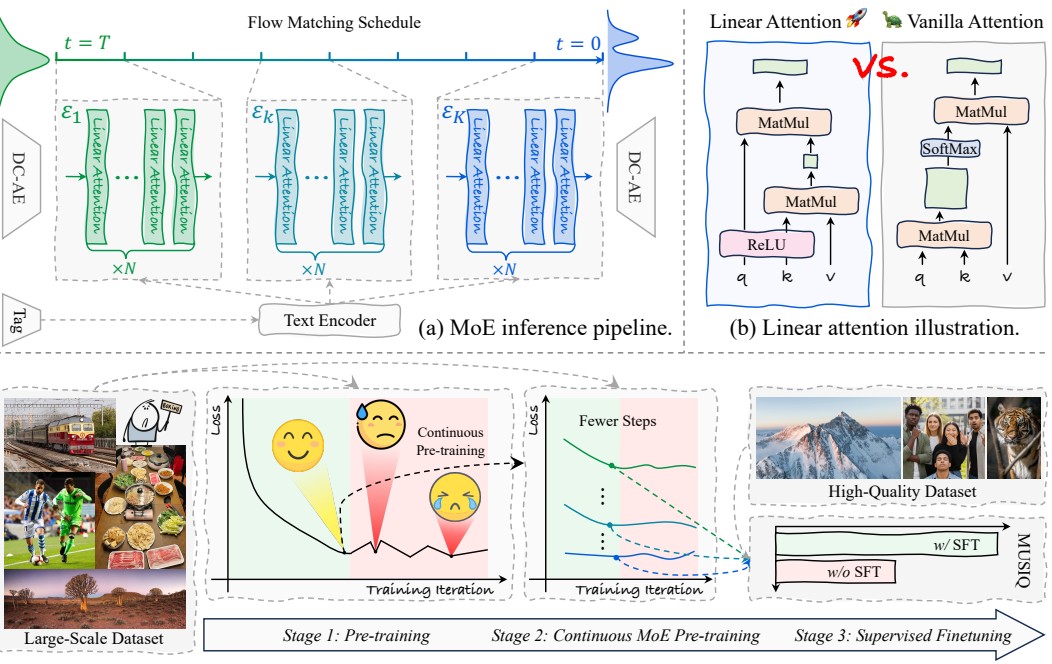

Figure 2: **The Integrated LinearSR Framework.** This figure illustrates how our contributions synergize: the tag-guided Mixture of Experts (MoE) architecture (a), built upon an efficient linear attention backbone (b), is made stable and effective by our Early-Stopping Guided Fine-tuning (ESGF) strategy (c), which initiates fine-tuning at the critical "knee point" to maximize performance.

plagued by training instability and a severe perception-distortion trade-off. Our work aims to bridge this critical gap, demonstrating the first successful integration of linear attention for high-fidelity diffusion-based super-resolution. A more detailed literature review is provided in Appendix B.

## 3 METHOD

LinearSR, our framework designed to dismantle the long-standing trade-off between computational efficiency and generative fidelity in super-resolution. Motivated by the need for a robust and practical $O(N)$ solution, we architected a system that seamlessly integrates a lightweight guidance paradigm with a stable, multi-stage training methodology. The synergy of these components, summarized in Fig. 2, establishes the first effective application of linear attention in high-fidelity SR.

### 3.1 LINEARSR FRAMEWORK

LinearSR is a conditional Diffusion Transformer (DiT) whose architecture is shown in Fig. 2(a). Its core is a DiT backbone using a ReLU-based Linear Attention, a mechanism validated for efficiency in high-resolution domains like dense prediction (Wang et al., 2020; Shen et al., 2021; Cai et al., 2022) and generative synthesis (Xie et al., 2024). Our work, for the first time, adapts this established architecture for the distinct challenges of high-fidelity generative super-resolution.

Standard self-attention computes a pairwise similarity matrix with $O(N^2)$ complexity. As contrasted in Fig. 2(b), linear attention avoids this bottleneck using the associative property of matrix multiplication. Given query, key, and value vectors $\mathbf{q}_i, \mathbf{k}_j, \mathbf{v}_j \in \mathbb{R}^d$, the output $\mathbf{o}_i$ is:

$$\mathbf{o}_i = \frac{\phi(\mathbf{q}_i)\left(\sum_{j=1}^{N} \phi(\mathbf{k}_j)^T \mathbf{v}_j\right)}{\phi(\mathbf{q}_i)\left(\sum_{j=1}^{N} \phi(\mathbf{k}_j)^T\right)} \tag{1}$$

where $\phi(\cdot) = \text{ReLU}(\cdot)$. Instead of associating each query $\mathbf{q}_i$ with every key $\mathbf{k}_j$, the operations are reordered. The terms $\sum \phi(\mathbf{k}_j)^T \mathbf{v}_j$ and its normalizer are computed first, forming a global summary

in a fixed-size tensor. Each query $\mathbf{q}_i$ then interacts with this pre-computed context, reducing the overall complexity to $O(N)$. To enhance performance, our backbone pairs linear attention with a Mix-FFN module. This module uses a $3 \times 3$ depth-wise convolution to bolster local information processing–compensating for a known weakness of linear attention–and accelerate convergence.

A critical adaptation for SR is injecting the low-resolution (LR) image condition. We introduce a lightweight conditioning stem, $\mathcal{E}_{conv}$, to process the LR input $x_{lr}$. This stem transforms $x_{lr}$ into a feature map with spatial dimensions matching the noisy latent $z_t$. The feature map is then concatenated with $z_t$ along the channel dimension to provide the DiT backbone with structural guidance, expressed as:

$$z'_t = \text{Concat}\left(z_t, \mathcal{E}_{conv}(x_{lr})\right) \tag{2}$$

The $\mathcal{E}_{conv}$ stem consists of three strided convolutional layers with SiLU activations. This learnable, multi-scale approach captures salient structural and content information from the LR image, offering superior guidance compared to fixed, non-learned upsampling techniques like bilinear interpolation.

## 3.2 GUIDANCE: A "PRECISION-OVER-VOLUME" APPROACH

Let our model be a vector field prediction network $v_\theta(z_t, t, c)$ trained with the Conditional Flow Matching (CFM) objective (Lipman et al., 2022; Tong et al., 2023):

$$\mathcal{L}_{\text{CFM}} = \mathbb{E}_{t, z_1 \sim q(z), z_0 \sim p_0(z)} \left[ \|(z_1 - z_0) - v_\theta((1-t)z_0 + tz_1, t, c)\|^2 \right] \tag{3}$$

where $z_1$ is a sample from the data distribution and $z_0$ is sampled from a prior, typically $\mathcal{N}(0, I)$. The model learns to approximate the vector field of a probability path that transports samples from the prior to the data distribution. A pivotal design choice for super-resolution (SR) is the nature of the conditioning context $c$. Unlike text-to-image synthesis, which creates content from external prompts, SR begins with a strong visual prior: the low-resolution (LR) image itself. This raises a fundamental question: is it more effective to supplement the model with rich, external descriptions, or to guide it by instead precisely extracting features already inherent to the LR input?

To investigate this, we explore two distinct guidance paradigms. The first is external semantic guidance using descriptive captions. The second, termed self-contained feature guidance, extracts features directly from the LR image. We evaluate a spectrum of models for this: at one end is CLIP (Radford et al., 2021), whose features are aligned with language concepts via contrastive learning. At the other is DINO (Caron et al., 2021), a self-supervised model that learns purely visual representations without textual supervision. Through self-distillation on augmented views, it is forced to learn features for object parts and structures, often unaligned with linguistic semantics. Bridging these two extremes, we consider a tag-style model inspired by SeeSR (Wu et al., 2024b), which uses a tagger like RAM (Zhang et al., 2024) to extract a concise set of object labels.

Previous work (Wu et al., 2024a;b) has shown tag-based guidance surpasses long-text captions. Our investigation expands on this by directly comparing it against powerful, vision-only models like CLIP and DINO. Our empirical findings, detailed in Sec. 4.3.1, revealed that interestingly, both DINO and CLIP features outperformed descriptive text. This suggests the core challenge in SR is not an information deficit but its effective utilization; adding external context is less effective than precisely extracting intrinsic semantics from the LR image. The tag-style model, providing a structured object vocabulary, yielded the best results, validating our "precision-over-volume" principle: a smaller, targeted guidance signal is indeed more effective and efficient for the SR task.

## 3.3 EARLY-STOPPING GUIDED FINE-TUNING (ESGF) FOR STABILITY

Fine-tuning the linear attention model for high-fidelity SR presented a challenge: the training invariably collapsed. We hypothesized this instability stemmed from the model converging to a sharp minimum in the loss landscape, known to cause poor generalization and adaptation instability (Keskar et al., 2016). In this state, the model over-specializes on artifacts instead of learning robust features.

To validate this critical hypothesis, we extensively analyzed the training dynamics. By tracking validation metrics against the ever-decreasing training loss, we discovered a universal pattern (Fig. 3(b)): performance metrics would improve, plateau, and then begin to erratically oscillate, proving conclusively that relying on loss alone is a deceptive guide for model selection. This observation led us to define the "knee-point" as the iteration of optimal generalization before performance degrades.

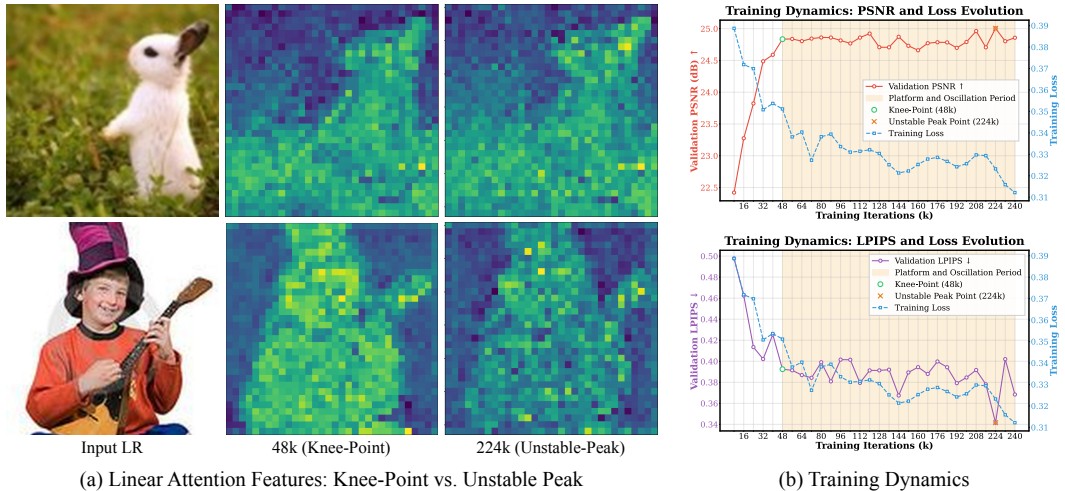

(a) Linear Attention Features: Knee-Point vs. Unstable Peak

(b) Training Dynamics

Figure 3: **Justification for ESGF through Instability Analysis.** (a) Representative feature maps from the same linear attention layer reveal a stark structural degradation from the knee-point to a later unstable peak. (b) The training dynamics confirm this phenomenon is universal, with PSNR and LPIPS metrics exhibiting the characteristic "Plateau and Oscillation Phase" post-knee-point.

As definitive proof, we compared internal model states. We visualized feature maps from the same linear attention layer, contrasting the knee-point model with one from a later "unstable peak" (Fig. 3(a)). The results are striking: features from the knee-point are structurally coherent, while those from the unstable peak are noisy and degraded, confirming a catastrophic loss of representational quality. Based on this evidence, we propose Early-Stopping Guided Fine-tuning (ESGF): all fine-tuning must initialize from the knee-point checkpoint. This model, residing in a flatter, more robust region of the loss landscape, provides a stable foundation for adaptation.

This theoretically-grounded strategy resolves the training collapse. This instability was a persistent bottleneck that suggested linear attention was fundamentally ill-suited for multi-stage SR training. Therefore, ESGF is not merely an enabler but a core innovation that makes our framework viable.

## 3.4 SNR-BASED MIXTURE OF EXPERTS FOR PERCEPTION-DISTORTION TRADE-OFF

A final obstacle to top-tier performance was the perception-distortion trade-off, where improving perceptual realism sacrificed fidelity. Our insight is that this trade-off is dynamic: early, high-noise stages (low SNR) demand coarse structure generation, while later, low-noise stages (high SNR) require detail refinement. To address this, we introduce an SNR-based Mixture of Experts (MoE) architecture.

Our approach, visualized in Fig. 4, partitions the generative trajectory within the log-Signal-to-Noise Ratio (log-SNR) space, $\lambda(t)$. The process operates over an effective range $[\lambda_{\min}, \lambda_{\max}]$ dictated by the noise schedule; for instance, the "scaled_linear" variant yields a characteristic asymmetric range (Diffusion, 2024). The partitioning is hierarchical: first, a primary anchor $\lambda_{\text{anchor}}$ at $t_2$ bisects the range

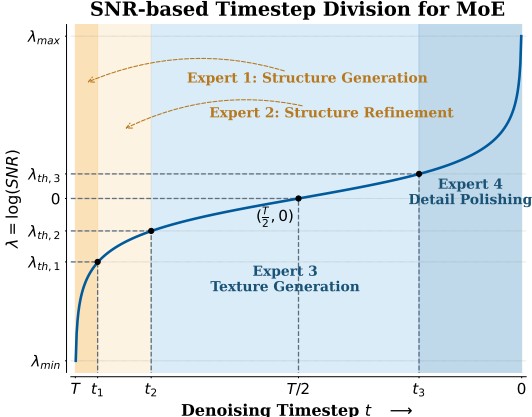

Figure 4: **Hierarchical log-SNR bisection defines operational boundaries for 4-expert MoE.**

into high-noise (structure) and low-noise (refinement) regimes, a division supported by related work (Wan-Video, 2025). We then further bisect these two sub-intervals in the log-SNR space, mapping their midpoints back to the time domain via $t(\lambda)$ to derive the final boundaries $t_1$ and $t_3$.

This hierarchical derivation yields the time boundaries $\{t_1, t_2, t_3\}$ for our four experts, $\{\mathcal{E}_k\}_{k=1}^4$ (detailed in App. A). A gating network uses these boundaries to deterministically route inputs, enabling specialized processing without inference overhead, as only one expert is active per timestep.

## 4 EXPERIMENTS

### 4.1 EXPERIMENTAL SETTINGS

**Training and Implementation Details.** For the pre-training and MoE stages, we use the public datasets DIV2K (Agustsson & Timofte, 2017), LSDIR (Li et al., 2023), and ReLAION-High-Resolution (laion, 2024), along with a custom high-resolution dataset crawled from Unsplash. The final SFT stage employs a curated set of high-quality images collected from the Internet. We synthesize 4x LR-HR pairs (256×256 from 1024×1024) using the Real-ESRGAN (Wang et al., 2021) degradation pipeline. Our model is trained with a flow-based objective (Lipman et al., 2022), employing the CAME optimizer (Luo et al., 2023) with a constant learning rate of 1e-4 and a batch size of 14. Stage 1 utilized 8 NVIDIA A800 GPUs, while each of the four experts in Stage 2 was trained on 6 A800 GPUs. Following our ESGF strategy, each stage proceeds until its performance 'knee-point' is reached. For inference, we use 20 sampling steps to generate results.

**Evaluation Datasets and Compared Methods.** We conduct evaluations on RealSR (Cai et al., 2019), DrealSR (Wei et al., 2020), RealLQ250 (Ai et al., 2024), and a synthetic test set of 100 images from DIV2K-Val (Agustsson & Timofte, 2017) generated with our training degradation. For RealSR and DrealSR, we follow the established protocol (Wu et al., 2024a;b; Chen et al., 2025a) of center-cropping inputs to 256×256. We compare LinearSR against ten state-of-the-art methods: StableSR (Wang et al., 2024a), DiffBIR (Lin et al., 2024), SeeSR (Wu et al., 2024b), SUPIR (Yu et al., 2024), DreamClear (Ai et al., 2024), SinSR (Wang et al., 2024b), OSEDiff (Wu et al., 2024a), AdcSR (Chen et al., 2025a), InvSR (Yue et al., 2025), and TSD-SR (Dong et al., 2025).

**Evaluation Metrics.** Following (Yu et al., 2024), we use PSNR, SSIM (Wang et al., 2004), and LPIPS (Zhang et al., 2018a) as full-reference metrics and MANIQA (Yang et al., 2022), CLIPIQA (Wang et al., 2023), and MUSIQ (Ke et al., 2021) as non-reference metrics for comprehensive evaluation. The former evaluate pixel-level fidelity, while the latter are designed to align with human perception. Our method, like other generative SR approaches (Ai et al., 2024; Lin et al., 2024; Wang et al., 2024a; Yu et al., 2024; Chen et al., 2025a; Yue et al., 2025; Dong et al., 2025), achieves strong results in non-reference metrics but performs less competitively on full-reference metrics.

### 4.2 COMPARISON WITH STATE-OF-THE-ARTS

Table 1: **Quantitative comparison with SOTA methods.** Best and second-best are highlighted.

| Datasets | Metrics | StableSR | DiffBIR | SeeSR | SUPIR | DreamClear | SinSR | OSEDiff | AdcSR | InvSR | TSD-SR | LinearSR |
|---|---|---|---|---|---|---|---|---|---|---|---|---|
| DIV2K-Val | PSNR↑ | 26.329 | 26.480 | 26.180 | 25.179 | 25.486 | 26.098 | 25.724 | 25.782 | 25.481 | 24.199 | 25.262 |
| | SSIM↑ | 0.646 | 0.680 | 0.711 | 0.656 | 0.658 | 0.634 | 0.688 | 0.674 | 0.695 | 0.621 | 0.684 |
| | LPIPS↓ | 0.421 | 0.443 | 0.374 | 0.426 | 0.397 | 0.526 | 0.396 | 0.397 | 0.426 | 0.408 | 0.401 |
| | MANIQA↑ | 0.281 | 0.474 | 0.473 | 0.400 | 0.376 | 0.393 | 0.429 | 0.403 | 0.429 | 0.438 | 0.475 |
| | MUSIQ↑ | 52.401 | 64.131 | 68.356 | 63.593 | 60.304 | 60.296 | 66.761 | 66.168 | 65.455 | 69.277 | 69.466 |
| | CLIPIQA↑ | 0.487 | 0.670 | 0.682 | 0.563 | 0.609 | 0.668 | 0.646 | 0.636 | 0.675 | 0.686 | 0.683 |
| RealSR | PSNR↑ | 25.346 | 25.008 | 25.702 | 24.103 | 23.907 | 25.982 | 24.754 | 25.183 | 24.299 | 23.736 | 23.838 |
| | SSIM↑ | 0.738 | 0.681 | 0.751 | 0.688 | 0.696 | 0.727 | 0.737 | 0.737 | 0.730 | 0.711 | 0.696 |
| | LPIPS↓ | 0.272 | 0.335 | 0.267 | 0.340 | 0.312 | 0.350 | 0.280 | 0.280 | 0.271 | 0.265 | 0.313 |
| | MANIQA↑ | 0.372 | 0.534 | 0.519 | 0.409 | 0.471 | 0.400 | 0.484 | 0.508 | 0.445 | 0.493 | 0.528 |
| | MUSIQ↑ | 63.352 | 67.241 | 69.254 | 63.302 | 65.213 | 59.313 | 69.738 | 70.505 | 68.670 | 70.493 | 70.552 |
| | CLIPIQA↑ | 0.561 | 0.690 | 0.686 | 0.515 | 0.691 | 0.653 | 0.682 | 0.695 | 0.681 | 0.723 | 0.673 |
| DrealSR | PSNR↑ | 25.758 | 25.158 | 26.212 | 24.835 | 25.186 | 25.734 | 25.455 | 25.768 | 24.483 | 24.264 | 25.235 |
| | SSIM↑ | 0.675 | 0.636 | 0.745 | 0.700 | 0.683 | 0.661 | 0.739 | 0.730 | 0.693 | 0.681 | 0.719 |
| | LPIPS↓ | 0.308 | 0.444 | 0.320 | 0.375 | 0.363 | 0.476 | 0.320 | 0.326 | 0.364 | 0.331 | 0.510 |
| | MANIQA↑ | 0.319 | 0.502 | 0.495 | 0.403 | 0.350 | 0.390 | 0.475 | 0.495 | 0.461 | 0.469 | 0.510 |
| | MUSIQ↑ | 56.500 | 63.868 | 67.429 | 63.125 | 57.164 | 58.505 | 68.051 | 69.025 | 68.046 | 68.495 | 69.073 |
| | CLIPIQA↑ | 0.530 | 0.704 | 0.702 | 0.564 | 0.624 | 0.673 | 0.723 | 0.736 | 0.738 | 0.757 | 0.713 |
| RealLQ250 | MANIQA↑ | 0.289 | 0.496 | 0.502 | 0.393 | 0.450 | 0.421 | 0.433 | 0.450 | 0.421 | 0.470 | 0.515 |
| | MUSIQ↑ | 56.496 | 68.162 | 70.912 | 65.476 | 67.126 | 63.641 | 70.013 | 70.534 | 66.831 | 71.505 | 71.914 |
| | CLIPIQA↑ | 0.508 | 0.706 | 0.703 | 0.574 | 0.688 | 0.698 | 0.673 | 0.692 | 0.677 | 0.704 | 0.720 |

**Quantitative Analysis.** We perform a comprehensive quantitative evaluation against state-of-the-art (SOTA) methods, with results detailed in Tab. 1. While many models compete on traditional fi-

Table 2: **Efficiency comparison for 1024x1024 SR (tested on NVIDIA H-series GPUs).** Best, second, and third are highlighted. LinearSR's SOTA 1-NFE time validates its core efficiency.

| Metrics (↓) | StableSR | DiffBIR | SeeSR | SUPIR | DreamClear | SinSR | OSEDiff | AdcSR | InvSR | TSD-SR | LinearSR |
|---|---|---|---|---|---|---|---|---|---|---|---|
| 1 Image Inference Time (s) | 78.405 | 25.543 | 13.632 | 13.632 | 16.319 | 8.999 | 1.086 | 0.561 | 0.667 | 12.635 | 0.830 |
| 1 NFE Forward Time (s) | 0.428 | 0.499 | 0.273 | 2.662 | 1.873 | 0.929 | 0.150 | 0.046 | 0.613 | 9.434 | 0.036 |

delity metrics, LinearSR consistently demonstrates superior performance in no-reference perceptual quality, which more accurately reflects human visual assessment. On the challenging RealLQ250 benchmark, LinearSR achieves a clean sweep, ranking first across the board in MANIQA (0.515), MUSIQ (71.914), and CLIPIQA (0.720). This trend of perceptual dominance is consistent across all tested datasets, such as achieving top scores in MANIQA and MUSIQ on both DIV2K-Val and DrealSR. This proves our model's exceptional ability to generate realistic and aesthetically pleasing images, successfully translating its architectural innovations into state-of-the-art generative quality without compromising on a strong balance in reference-based metrics like LPIPS.

**Efficiency Analysis.** The core advantage of our work is validated in Tab. 2, which benchmarks computational efficiency. To ensure a fair and rigorous comparison, all measurements were conducted on the same GPU with no other tasks running. The reported times are the average over 100 runs, each processing a 256×256 input to a large-scale 1024×1024 output. Crucially, we focus on the 1-NFE (Number of Function Evaluations) forward time, a metric that isolates the performance of the core diffusion step by excluding the VAE decoder. LinearSR establishes a new state-of-the-art in this metric at just 0.036s for a 1024×1024 image in 1-NFE time, significantly outperforming previous methods that often benchmarked at smaller 512×512 resolutions. This highlights the architectural efficiency of our linear attention, especially when compared to one-step methods like TSD-SR, whose extensive time is attributable to its tile-based processing logic. This achievement is purely architectural; our method is orthogonal to, not mutually exclusive with, distillation techniques. This indicates that substantial room for further optimization exists by applying future distillation methods to our already efficient base model. While AdcSR and InvSR show faster overall inference due to model distillation and optimized sampling strategies respectively, LinearSR's total time of 0.830s remains highly competitive and is orders of magnitude faster than heavyweight models like SUPIR.

**Qualitative Analysis.** Beyond metrics, a qualitative comparison in Fig. 5 reveals the practical impact of our approach. While competing methods can produce plausible results, they often suffer from common generative pitfalls, such as introducing unnatural artifacts or an overly smooth, "painterly" effect that erases fine details. This dichotomy is evident, where some methods fail to resolve the initial degradation, leaving behind blur, while others sacrifice authentic detail for a smooth finish. In contrast, LinearSR excels at restoring crisp, realistic textures across diverse scenes. For example, in the case of the flower, our model reconstructs the delicate stamens with high clarity and preserves the subtle curvature and shadows of the petals, which are lost in other methods. Likewise, it renders the axolotl's eye with sharp definition and faithfully captures the fine, porous texture of its skin and the intricate details of its external gills. This qualitative edge is a direct result of our holistic framework, where the stable training enabled by ESGF and the specialized refinement from the SNR-based MoE effectively translate the efficiency of linear attention into superior visual fidelity.

## 4.3 ABLATION STUDY

To systematically validate our core contributions, we conducted a series of targeted ablations. This section dissects the impact of our guidance paradigm and the ESGF training strategy, demonstrating that they are integral components underpinning the performance and stability of LinearSR.

### 4.3.1 VALIDATION OF THE "PRECISION-OVER-VOLUME" GUIDANCE PRINCIPLE

We investigate different guidance methods, with results in Tab. 3. An interesting trend emerged: guidance from raw visual features (DINO, CLIP) significantly outperformed

Table 3: **Quantitative comparison of guidance methods.**

| Method | PSNR↑ | SSIM↑ | LPIPS↓ | MANIQA↑ | MUSIQ↑ | CLIPIQA↑ |
|---|---|---|---|---|---|---|
| Origin | 22.05 | 0.4267 | 0.6324 | 0.4541 | 60.10 | 0.6964 |
| CLIIP | 23.79 | 0.6270 | 0.4260 | 0.3510 | 60.75 | 0.5520 |
| DINO | 23.83 | 0.6560 | 0.3860 | 0.3370 | 62.76 | 0.5560 |
| **TAG** | 24.85 | 0.6910 | 0.3740 | 0.3630 | 63.93 | 0.5720 |

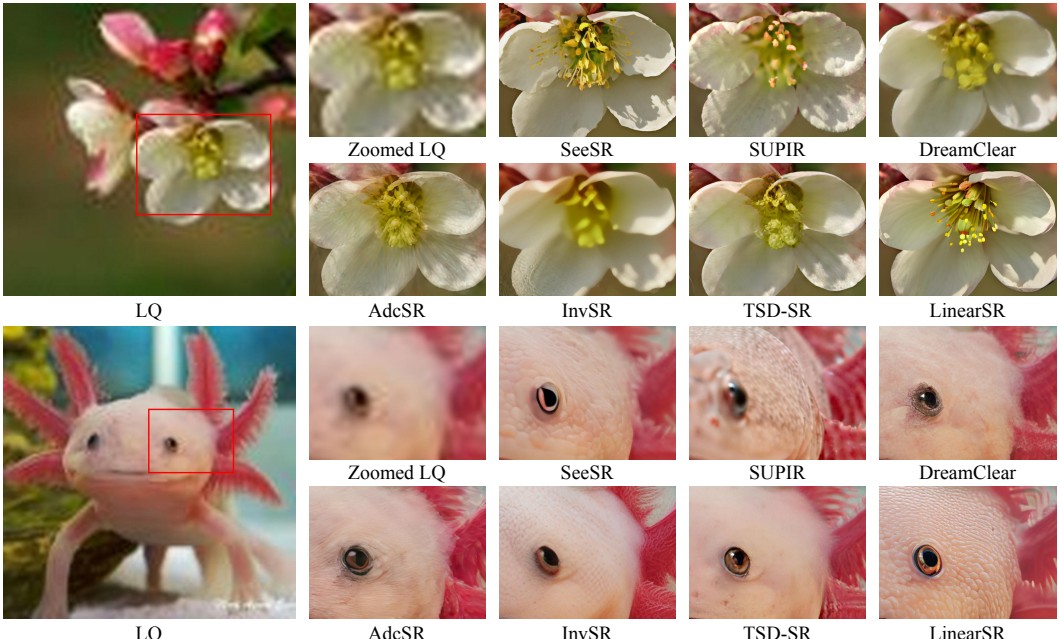

Figure 5: **Qualitative comparison with state-of-the-art methods.** Our LinearSR consistently restores intricate textures and realistic details, outperforming competing methods across diverse real-world degradations. This is particularly evident in its ability to reconstruct the flower's delicate stamens and petal textures, as well as the axolotl's complex skin pattern and sharp eye.

the Origin baseline using verbose sentence-level descriptions. This progression culminates with the **TAG** model, which provides concise object labels and is the definitive winner across nearly all critical metrics. This outcome validates our "precision-over-volume" principle, demonstrating that for the SR task, a concise, high-recall set of object labels is a more effective and efficient guidance signal. The qualitative evidence in Fig. 6(a) corroborates this, as the TAG-guided model is shown to clearly restore intricate details, such as the flower's stamens and previously illegible text.

Table 4: **Comparison of training strategies for the second stage.** Our strategy of selecting the checkpoint at the knee-point (48k) ensures stable training, whereas a naive selection from a seemingly optimal late-stage Unstable-Peak checkpoint (224k) inevitably leads to training collapse.

| Strategy | 1st Stage Checkpoint | 2nd Stage Training Status | PSNR↑ | SSIM↑ | LPIPS↓ | MANIQA↑ | MUSIQ↑ | CLIPIQA↑ |
|---|---|---|---|---|---|---|---|---|
| Naive Selection | 224k (Unstable-Peak) | Collapse (2k) | 23.59 | 0.664 | 0.403 | **0.459** | 60.39 | 0.663 |
| **Our Strategy** | 48k (Knee-Point) | **Stable (Completed)** | **24.78** | **0.667** | **0.410** | 0.452 | **64.59** | **0.690** |

### 4.3.2 NECESSITY OF THE ESGF STRATEGY FOR STABLE FINE-TUNING

Next, we demonstrate the critical role of our Early-Stopping Guided Fine-tuning (ESGF) strategy. We compare two approaches for initiating second-stage fine-tuning: a naive selection from a late-stage "Unstable-Peak" checkpoint versus our ESGF-guided selection from the "Knee-Point". As shown in Tab. 4, the outcome is decisive. This checkpoint selection is the decisive factor for training stability. The naive approach quickly leads to a training *Collapse*, yielding a poor final model. In contrast, our strategy ensures a *Stable* and complete fine-tuning process, resulting in a significantly better-performing model. This proves that ESGF is not merely an optimization but a foundational enabler, resolving the inherent instability of multi-stage training for linear attention SR models.

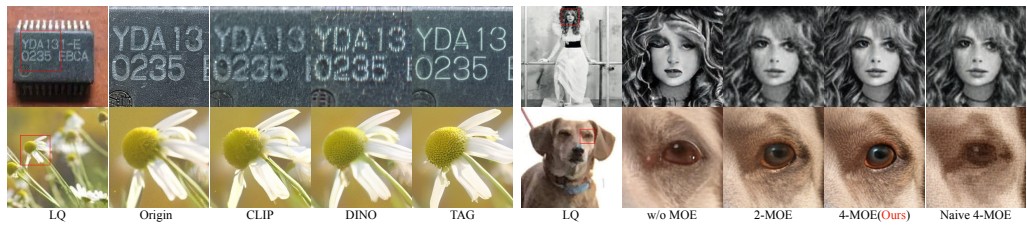

(a) Ablation on guidance methods                    (b) Ablation on MoE configurations

Figure 6: **Qualitative ablation study of our key components.** (a) Visual comparison of guidance methods, where our TAG-based approach, validating the "precision-over-volume" principle, restores superior texture and illegible details. (b) Visual comparison of MoE designs, demonstrating that our SNR-based 4-expert architecture yields the most realistic results by avoiding generative artifacts.

Table 5: **Ablation study on Mixture-of-Experts (MoE) configurations on DrealSR Dataset.**

| Exp. | Configuration | Partitioning Strategy | Boundaries (t) | PSNR↑ | SSIM↑ | LPIPS↓ | MANIQA↑ | MUSIQ↑ | CLIPIQA↑ |
|---|---|---|---|---|---|---|---|---|---|
| (a) | Baseline | N/A | N/A | 24.85 | 0.691 | 0.374 | 0.363 | 63.93 | 0.572 |
| (b) | 2-Expert MoE | SNR-based | [0.875] | 25.02 | 0.671 | 0.377 | 0.374 | 63.18 | 0.591 |
| (c)Ours | 4-Expert MoE | SNR-based | [0.223, 0.875,0.939] | 25.00 | 0.682 | 0.375 | 0.371 | 64.02 | 0.598 |
| (d) | 4-Expert MoE | Naive Uniform | [0.25, 0.5,0.75] | 24.84 | 0.666 | 0.389 | 0.368 | 62.51 | 0.582 |

### 4.3.3 EFFECTIVENESS OF THE SNR-BASED MoE ARCHITECTURE

We evaluate our SNR-based Mixture of Experts (MoE) architecture by first examining simpler approaches. As shown in Tab. 5 and Fig. 6(b), the model without MoE fails to generate fine details, while a naive uniform partitioning yields blurry and distorted results, failing to properly handle the distinct generative stages. This demonstrates that our core strategy of SNR-based expert specialization is essential for success. Building on this validated approach, our 4-expert model proves superior to a simpler 2-expert version. It renders visibly finer details in the woman's face and the dog's eye, ultimately achieving the highest perceptual scores and the best overall performance.

Table 6: **Progressive ablation study of our main contributions.** We demonstrate ESGF is a prerequisite for stable fine-tuning, as the naive approach (Exp. 3) results in training collapse.

| Exp. | TAG Prompt | ESGF | SNR-based 4-MoE | MoE SFT | PSNR↑ | SSIM↑ | LPIPS↓ | MANIQA↑ | MUSIQ↑ | CLIPIQA↑ |
|---|---|---|---|---|---|---|---|---|---|---|
| (1) Baseline | | | | | 22.05 | 0.427 | 0.632 | 0.454 | 60.10 | 0.696 |
| (2) Add Guidance | ✓ | | | | 24.85 | 0.691 | 0.374 | 0.363 | 63.93 | 0.572 |
| (3) Naive FT | ✓ | | ✓ | | *Training Collapse* | | | | | |
| (4) Add MoE | ✓ | ✓ | ✓ | | 25.00 | 0.682 | 0.375 | 0.371 | 64.02 | 0.598 |
| **LinearSR** | ✓ | ✓ | ✓ | ✓ | **25.24** | **0.719** | **0.359** | **0.510** | **69.07** | **0.713** |

### 4.3.4 PROGRESSIVE CONTRIBUTION OF COMPONENTS

Finally, we analyze the progressive contributions of our framework's components in Tab. 6. The first step, replacing original sentences with the TAG Prompt (Exp. 2 vs. 1), yields a dramatic boost across key fidelity metrics by improving guidance. Next, we introduce the SNR-based 4-MoE (Exp. 3 & 4), fine-tuned from the Exp. 2 checkpoint. It is critical to note that this stage is only made possible by leveraging ESGF to select a stable starting point. This intervention is crucial, as our attempts at direct fine-tuning proved unstable and would invariably collapse without this essential safeguarding mechanism. The final LinearSR model then applies the full, ESGF-guided two-stage MoE fine-tuning (MoE-SFT), pushing all metrics to their peak, especially in perceptual quality. This step-by-step analysis confirms that each component is indispensable and synergistic, with their interplay culminating in the superior performance of our complete framework.

## 5 CONCLUSION

In this work, we introduce LinearSR, the first framework to successfully unlock the potential of linear attention for high-fidelity super-resolution. By combining precision guidance, a specialized expert architecture, and a multi-stage, early-stopping fine-tuning strategy, we systematically dismantle the technical barriers that have historically hindered its application. Crucially, our architectural approach is orthogonal to, not mutually exclusive with, post-hoc optimizations like model distillation and pruning. By forging the first viable pathway for linear attention in the super-resolution domain, our work establishes a foundational and efficient baseline for future research to build upon.

## ACKNOWLEDGMENTS

This work is supported by Shanghai Artificial Intelligence Laboratory.

## ETHICS STATEMENT

Our LinearSR is the first framework to successfully unlock the potential of linear attention for high-fidelity super-resolution. To ensure ethical compliance, our training data was curated from a combination of established public datasets and publicly accessible online platforms, such as Unsplash. We have taken deliberate measures to minimize potential biases in this data, fully aligning with universal ethical guidelines. We explicitly emphasize that this framework is not intended for misuse in achieving harmful purposes; downstream users are encouraged to adhere to ethical principles when applying the technology. Additionally, all authors declare no conflicts of interest related to this work.

## REPRODUCIBILITY STATEMENT

To enable direct reproduction of the quantitative metrics in our tables and the visual comparisons in our figures, we plan to release the corresponding pre-trained model checkpoints and evaluation scripts, pending institutional approval. To ensure a consistent testing environment, we will also provide detailed specifications of the hardware used, such as GPU models, and the exact versions of all software libraries. This commitment ensures that our reported results can be precisely verified by the community.

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

# A   A HIERARCHICAL FRAMEWORK FOR MoE BOUNDARY DETERMINATION

In our main paper, we propose a Mixture-of-Experts (MoE) architecture to assign specialized sub-networks to distinct phases of the flow-matching-based generation process. The efficacy of this approach is contingent upon a principled partitioning of the continuous time variable $t \in [0, 1]$. This section provides a detailed, step-by-step derivation of the boundaries for our 4-expert model, rooted in a hierarchical partitioning of the log-Signal-to-Noise Ratio (log-SNR) space.

Our framework begins with the log-SNR definition for our Flow Matching model, $\lambda(t)$, and its inverse, $t(\lambda)$:

$$\lambda(t) = 2(\log(1 - t) - \log(t)) \tag{4}$$

$$t(\lambda) = \frac{1}{\exp\left(\frac{\lambda}{2}\right) + 1} \tag{5}$$

## A.1   DERIVATION OF THE EFFECTIVE LOG-SNR RANGE

The operational domain of our model is defined by an effective log-SNR range, $[\lambda_{\min}, \lambda_{\max}]$. This range is not arbitrary but is derived from the noise schedule of foundational models like Stable Diffusion 1.5 (Diffusion, 2024), whose practices we adopt. Specifically, the log-SNR is directly related to the noise level $\sigma$ in many diffusion frameworks by:

$$\lambda = -2\log(\sigma) \tag{6}$$

The Stable Diffusion 1.5 repository and its common implementations (e.g., in k-diffusion) define an effective noise schedule bounded by $\sigma_{\min}$ and $\sigma_{\max}$. For our model's configuration, the effective boundaries correspond to $\sigma_{\min\_eff} \approx 0.0118$ and $\sigma_{\max\_eff} \approx 33.78$. Plugging these into the equation yields our operational range:

$$\lambda_{\max} = -2\log(\sigma_{\min\_eff}) = -2\log(0.0118) \approx 8.87 \tag{7}$$

$$\lambda_{\min} = -2\log(\sigma_{\max\_eff}) = -2\log(33.78) \approx -7.04 \tag{8}$$

Thus, our effective log-SNR range is established as $[\lambda_{\min}, \lambda_{\max}] \approx [-7.04, 8.87]$.

## A.2   FOUNDATIONAL BISECTION: THE 2-EXPERT CONCEPTUAL MODEL

The most fundamental division of labor in the generative process is separating the high-noise regime (where global structure is formed) from the low-noise regime (where details are refined). This motivates a conceptual 2-expert model. This principle of partitioning the generation process based on noise levels has been shown to be highly effective. For instance, eDiff-I (Balaji et al., 2022) successfully employed an ensemble of specialized denoisers, one for high-noise steps and another for low-noise steps, validating the benefit of such a separation.

Inspired by this, we establish a logical boundary between these two regimes, which can be conceptualized as a "perceptual turning point" where the model's output transitions from being noise-dominated to structure-dominated. We instantiate this idea by selecting a concrete anchor point, $t_{anchor} = 0.875$. This value represents a state where the coarse structure is largely formed, but fine details are still absent. We define its corresponding log-SNR value as our primary anchor, $\lambda_{anchor}$:

$$\lambda_{anchor} = \lambda(t_{anchor}) = \lambda(0.875) \approx -3.89 \tag{9}$$

This anchor point partitions the effective log-SNR range into two primary operational zones:

- **High-Noise Zone (Structure Formation):** $\lambda \in [-7.04, -3.89]$
- **Low-Noise Zone (Detail Refinement):** $\lambda \in [-3.89, 8.87]$

## A.3   REFINED QUADRISECTION: EXTENDING TO A 4-EXPERT ARCHITECTURE

To achieve finer-grained specialization, we extend the 2-expert model to a 4-expert architecture by further partitioning each of the two primary zones. To ensure that the task complexity is distributed equitably among the new experts, we bisect each log-SNR sub-interval at its midpoint. This midpoint bisection strategy, also empirically validated in related work (Wan-Video, 2025), introduces two secondary boundaries, $\lambda_1$ and $\lambda_3$:

1. **Low-Noise Zone Bisection ($\lambda_1$):** To separate detail-refinement from texture-generation, we define $\lambda_1$ as the midpoint of the low-noise interval $[\lambda_{\text{anchor}}, \lambda_{\text{max}}]$:

$$\lambda_1 = \frac{\lambda_{\text{anchor}} + \lambda_{\text{max}}}{2} = \frac{-3.89 + 8.87}{2} \approx 2.49 \tag{10}$$

2. **High-Noise Zone Bisection ($\lambda_3$):** To distinguish between initial denoising and coarse structure formation, we define $\lambda_3$ as the midpoint of the high-noise interval $[\lambda_{\text{min}}, \lambda_{\text{anchor}}]$:

$$\lambda_3 = \frac{\lambda_{\text{min}} + \lambda_{\text{anchor}}}{2} = \frac{-7.04 + (-3.89)}{2} \approx -5.47 \tag{11}$$

## A.4 FINAL EXPERT BOUNDARIES

The three calculated log-SNR values–$\lambda_1 = 2.49$, $\lambda_{\text{anchor}} = -3.89$ (renamed to $\lambda_2$ for consistency), and $\lambda_3 = -5.47$–serve as the final boundaries for our 4-expert model. We use the inverse function $t(\lambda)$ (Eq. 5) to map these back to the time domain, yielding the final partitions summarized in Table 7.

Table 7: Final MoE Boundaries in log-SNR and Time Domains, ordered by generation process.

| Expert | Primary Task | log-SNR Range | Time ($t$) Range |
|--------|-------------|---------------|------------------|
| Expert 1 | Initial Denoising | $[-7.04, -5.47]$ | $[0.939, 1.0]$ |
| Expert 2 | Coarse Structure | $[-5.47, -3.89]$ | $[0.875, 0.939]$ |
| Expert 3 | Texture Generation | $[-3.89, 2.49]$ | $[0.223, 0.875]$ |
| Expert 4 | Detail Refinement | $[2.49, 8.87]$ | $[0.0, 0.223]$ |

# B DETAILED RELATED WORK

## B.1 IMAGE RESTORATION WITH DIFFUSION MODELS

Recent years have witnessed a paradigm shift in image restoration, moving from traditional methods based on Convolutional Networks (Dong et al., 2015; 2014; 2016; Zhang et al., 2018b) and Transformers (Chen et al., 2021; Liang et al., 2021; Zamir et al., 2022; Chen et al., 2025b) to approaches that leverage large-scale generative models as powerful priors. The emergence of diffusion models has been particularly transformative, enabling the generation of visually realistic and semantically consistent outputs even under severe degradation. A prevailing trend has been to build increasingly large models to tackle blind restoration, exemplified by works like DiffBIR (Lin et al., 2024) and SeeSR (Wu et al., 2024b). This paradigm culminated in models like SUPIR (Yu et al., 2024), which leverages the powerful SDXL prior to achieve an exceptional balance between perceptual quality and fidelity, and DreamClear (Ai et al., 2024), the first to apply a pure Diffusion Transformer (DiT) architecture directly to super-resolution.

As these large-scale models demonstrated state-of-the-art capabilities, their prohibitive inference costs became a major bottleneck. Consequently, the research focus began to shift towards acceleration. Subsequent works have pursued this direction by employing techniques such as knowledge distillation, model compression, and diffusion inversion to significantly reduce inference costs while maintaining high performance (Wang et al., 2024b; Wu et al., 2024a; Dong et al., 2025; Yue et al., 2025; Chen et al., 2025a). However, a common thread in these models is their heavy reliance on the standard self-attention mechanism. The quadratic computational complexity of this mechanism becomes a severe bottleneck as input resolution increases, motivating the exploration of more efficient alternatives.

## B.2 LINEAR ATTENTION

To address the quadratic complexity ($O(N^2)$) of standard self-attention, linear attention methods were developed to reduce the computational cost to $O(N)$. The core idea, shared across many variants, is to exploit the associative property of matrix multiplication. Instead of explicitly computing

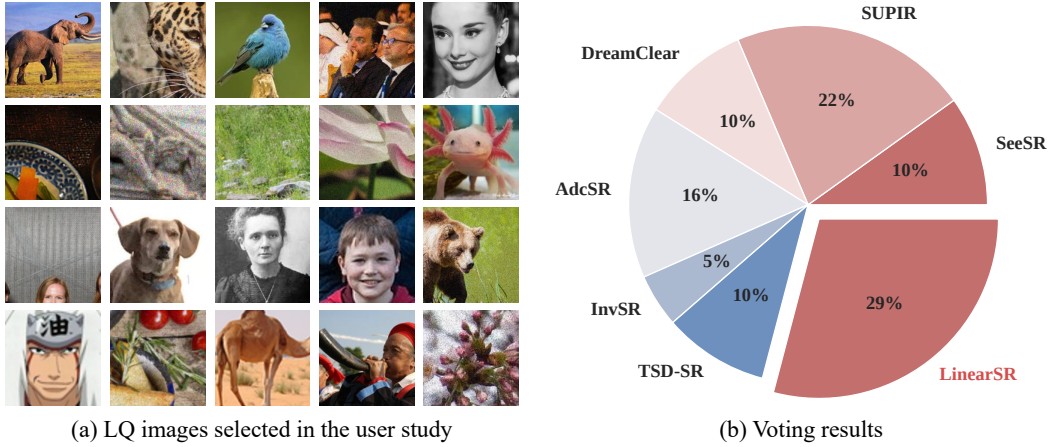

(a) LQ images selected in the user study      (b) Voting results

Figure 7: User study on perceptual preference. (a) The 20 LQ images used for evaluation. (b) Score proportion across methods from 50 participants with Top-3 voting (3/2/1).

the $N \times N$ attention matrix in $(QK^T)V$, these methods reorder the computation to $Q(K^T V)$, thus avoiding the expensive quadratic term (Shen et al., 2021). This concept was pioneered in the NLP domain with methods like Linformer (Wang et al., 2020), which used low-rank projections, and the work of Katharopoulos et al. (Katharopoulos et al., 2020), which introduced kernel-based feature maps to linearize the attention calculation.

Building on these foundations, linear attention has been successfully extended into the computer vision domain, proving particularly effective for high-resolution tasks. For instance, Efficient Attention (Shen et al., 2021) improved performance on object detection, while EfficientViT (Cai et al., 2022) achieved state-of-the-art efficiency in dense prediction tasks like semantic segmentation. More recently, the application of linear attention has expanded into generative modeling, where Sana (Xie et al., 2024) demonstrated its effectiveness for efficient, high-resolution text-to-image synthesis within a Diffusion Transformer (DiT). The work establishes linear attention as a mature and effective technique for reducing the computational burden of Transformers, providing a strong foundation for our work.

## C  USER STUDY

To comprehensively evaluate the perceptual quality and content fidelity of our method, we conducted a controlled user study, following the established practices of recent Real-ISR works (Dong et al., 2025; Chen et al., 2025a; Wu et al., 2024a; Ai et al., 2024).

**Experimental Setup.**  We curated a diverse test set of 20 low-quality (LQ) images, which are shown in Figure 7(a). This set was specifically designed to cover different degradation types, comprising 3 synthetically degraded images from the DIV2K validation set and 17 real-world degraded images from the RealLQ250 dataset. We compared our method against six strong diffusion-based baselines: TSD-SR, InvSR, AdcSR, DreamClear, SUPIR, and SeeSR.

**Evaluation Protocol.**  We recruited 50 participants for the study. For each of the 20 source images, participants were presented with the results from all seven methods in a randomized order, alongside the original LQ input for reference. The presentation order was reshuffled for each participant and each image to prevent bias. Participants were instructed to evaluate the results based on two equally weighted criteria: (1) overall perceptual quality (e.g., clarity, detail, and realism) and (2) content consistency with the input (i.e., faithful restoration of structure and texture).

**Scoring Mechanism.**  We employed a Top-3 voting scheme. For each source image, participants selected their top three preferred results, assigning 3 points to their first choice, 2 points to the second, and 1 point to the third. This means each participant distributed 6 points per image. The total points for each method $m$ across all participants and images, denoted as $S_m$, were aggregated.

To quantify the overall preference, we calculated the final score proportion $p_m$ as:

$$p_m = \frac{S_m}{\sum_k S_k}$$

With 50 participants, 20 images, and 6 points awarded per image, a total of 6,000 points were distributed among the methods. This score proportion, representing a method's relative share of user preference, is visualized in Figure 7(b). As shown, our method achieves the highest score proportion, indicating a strong user preference for its superior restoration quality.

## D  ROBUSTNESS OF THE KNEE-POINT SELECTION STRATEGY

In this section, we elaborate on the rigorous methodology used to determine the optimal training termination step, referred to as the "Knee-Point." Rather than relying on arbitrary heuristics or manual guesswork, we employ a systematic two-stage strategy: (1) Automated Detection via Metric Variance Analysis, followed by (2) Human Verification.

### D.1  AUTOMATED DETECTION ALGORITHM

To identify the precise transition from the convergence phase to the oscillation phase, we track 6 metrics (PSNR, SSIM, LPIPS, MANIQA, MUSIQ, CLIPIQA) across 4 diverse datasets (DIV2K-Val, RealSR, DrealSR, RealLQ250). This covers both synthetic and real-world distributions.

We define the Knee-Point $t^*$ as the critical step where the model achieves maximum stability before performance degradation begins. The detection process is formalized in **Algorithm 1**. Let $\mathcal{M} = \{m_1, m_2, \ldots, m_T\}$ be the sequence of a validation metric recorded at steps $t$. We utilize a sliding window of size $W$ to monitor the local variance and the future trend slope.

---

**Algorithm 1** Automated Knee-Point Detection Strategy

---

**Require:** Metric sequence $\mathcal{M} = \{m_t\}_{t=1}^{T}$, Window size $W$, Stability threshold $\epsilon_{stable}$
**Ensure:** Optimal Knee-Point $t^*$
1: Initialize candidates set $\mathcal{C} \leftarrow \emptyset$
2: **for** $t = W$ **to** $T - W$ **do**
3:     {Calculate local variance over the past window}
4:     $V_t \leftarrow \text{Var}(m_{t-W:t})$
5:     {Calculate slope trend over the future window}
6:     $S_t \leftarrow \text{Slope}(m_{t:t+W})$
7:     {Identify stability region before negative trend}
8:     **if** $V_t < \epsilon_{stable}$ **and** $S_t < 0$ **then**
9:         $\mathcal{C} \leftarrow \mathcal{C} \cup \{t\}$
10:     **end if**
11: **end for**
12: {Select the latest step satisfying criteria}
13: $t^* \leftarrow \max(\mathcal{C})$
14: **return** $t^*$

---

Following the automated proposal of $t^*$, we conduct a Human Verification phase. We visually inspect the validation curves and generated samples around $t^*$ to ensure the detected point does not correspond to a temporary local fluctuation but represents a genuine structural convergence.

### D.2  EMPIRICAL VALIDATION AND UNIVERSALITY

To demonstrate the universality of this phenomenon, we present the training log of a separate optimization experiment in **Table 8** and visualize the trend in **Figure 8**.

While the Knee-Point in our main paper was detected at **48k** steps, this separate experimental run exhibits the Knee-Point at **58k** steps. Despite the shift in the absolute step number due to different hyperparameter settings, the underlying pattern remains consistent:

1. **Rapid Ascent Phase:** Metrics improve significantly in the early stages.

2. **Knee-Point ($t^*$):** Performance peaks and variance is minimized (highlighted in bold in Table 8).

3. **Oscillation Phase:** Beyond $t^*$, the metrics enter a phase of high variance or slight degradation, indicating overfitting or instability inherent to the adversarial/perceptual loss components.

As shown in Table 8, at step 58k, the model achieves the optimal trade-off across perception (MANIQA, MUSIQ) and fidelity (PSNR, SSIM) metrics. For instance, on the RealSR dataset, MUSIQ peaks at 60.231 at 58k before dropping to 51.643 at 242k, confirming the necessity of our early stopping strategy.

## E    BENCHMARKING PROTOCOLS AND QUALITY-LATENCY ALIGNMENT

In this section, we provide a detailed clarification of our benchmarking methodology, specifically focusing on the measurement of inference latency and the structural differences between the original and distilled models.

### E.1    DEFINITION OF 1-NFE FORWARD TIME

The reported "1-NFE Forward Time" in our experiments strictly measures the **core diffusion denoising step**. This measurement excludes auxiliary computational costs such as the Text Encoder, VAE decoding, and pre-processing steps. This isolation ensures that the metric purely reflects the architectural efficiency of the diffusion backbone itself.

### E.2    THE "QUALITY-LATENCY ALIGNMENT" STANDARD

To ensure a rigorous and fair comparison, we adhere to a **Quality-Latency Alignment** standard. This principle dictates that the latency reported for a model must be measured using the exact configuration (e.g., hyperparameters, guidance strategies) required to achieve the optimal visual quality presented in the qualitative comparisons.

**Impact of Classifier-Free Guidance (CFG).**    A critical factor influencing this measurement is the use of Classifier-Free Guidance (CFG). Based on this principle, the "unit work" per step differs across methods:

- **Methods with CFG (Heavier):** Like LinearSR, DiffBIR, DreamClear, SeeSR, and SUPIR, these models default to using CFG. Their "core denoising step" physically necessitates computing both conditional and unconditional branches (effective batch size = 2).

- **Methods without CFG (Lighter):** Distilled or single-branch methods like AdcSR, OSEDiff, InvSR, TSD-SR, and SinSR typically use a single branch (effective batch size = 1).

### E.3    EFFICIENCY ANALYSIS

It is acknowledged that single-branch models naturally possess a computational advantage in 1-NFE time due to their reduced workload. Despite this structural difference, our model achieves a state-of-the-art 1-NFE efficiency of 0.036s. This performance highlights the intrinsic speed of LinearSR. It indicates that our architectural optimizations are orthogonal to distillation, suggesting significant potential for further acceleration when combined with such techniques.

## F    MORE VISUAL COMPARISONS

To further demonstrate the robustness and superiority of our proposed **LinearSR**, we provide extensive visual comparisons against state-of-the-art diffusion-based methods. Figures 9 and 10 showcase results on challenging real-world and synthetic degradation scenarios, respectively. These qualitative results comprehensively validate our method's effectiveness.

Across numerous examples, we observe several distinct patterns among the competing methods. Firstly, highly generative models like SUPIR and SeeSR, while capable of producing sharp details, often suffer from a tendency to hallucinate. They may introduce plausible but factually incorrect textures or even alter fine structures, sacrificing fidelity for perceptual sharpness. For instance, in several cases, they generate overly stylized patterns on fabrics or unnatural sheens on surfaces that are not present in the original content.

Secondly, method such as InvSR tends to be more conservative. While it generally maintains good structural consistency, it often yields over-smoothed results. This is particularly evident in its failure to restore high-frequency details, such as the intricate texture of sculpture, the fine strands of animal fur, or delicate patterns in foliage, leading to a less realistic and visually flat appearance.

Thirdly, other competitive methods like AdcSR, TSD-SR and DreamClear deliver strong performance but are not without flaws. They occasionally introduce subtle color shifts or minor artifacts, especially when dealing with complex textures or severe degradation, indicating a lack of robustness in the most challenging scenarios.

In stark contrast, our **LinearSR** consistently strikes an exceptional balance between fidelity and realism. As shown in Figure 9, on real-world degraded images, our method excels at restoring authentic and natural-looking details without introducing distracting artifacts. It successfully reconstructs crisp textures and clean edges where other methods struggle. Furthermore, as demonstrated in Figure 10, even under severe synthetic degradation, **LinearSR** shows remarkable fidelity to the ground truth. It faithfully recovers complex structures and avoids the distortions or blurring effects that plague other approaches. This robust performance across diverse and challenging conditions highlights the significant advantage of our method in producing perceptually pleasing and structurally coherent super-resolution results.

## G    DECLARATION OF USE OF LARGE LANGUAGE MODELS (LLM)

We affirm that this paper was primarily written by the authors. Large Language Models (LLMs) were utilized solely as general-purpose assistive tools for language refinement, grammar correction, and stylistic improvements during the writing process. Specifically, Gemini 2.5 Pro (Comanici et al., 2025) was employed for minor text polishing and rephrasing to enhance clarity and readability. No LLM was used for conceptual ideation, experimental design, data analysis, or generating any substantive content of the research.

Table 8: Full training log across all steps. The **58k** step is highlighted as the Knee-Point.

| Datasets | Metrics | 4k | 12k | 20k | 28k | 36k | 44k | 50k | 58k | 66k | 70k | 74k | 80k | 82k | 96k | 128k | 142k | 158k | 180k | 196k | 212k | 226k | 242k |
|---|---|---|---|---|---|---|---|---|---|---|---|---|---|---|---|---|---|---|---|---|---|---|---|
| DIV2K-Val | PSNR↑ | 21.204 | 23.017 | 23.476 | 24.211 | 24.604 | 24.619 | 24.862 | **24.778** | 25.302 | 24.943 | 24.767 | 25.251 | 24.648 | 24.969 | 24.813 | 24.802 | 25.029 | 25.165 | 25.159 | 25.324 | 25.306 | 25.155 |
| | SSIM↑ | 0.559 | 0.589 | 0.611 | 0.643 | 0.631 | 0.625 | 0.644 | **0.647** | 0.661 | 0.648 | 0.651 | 0.662 | 0.639 | 0.650 | 0.638 | 0.648 | 0.655 | 0.663 | 0.666 | 0.664 | 0.674 | 0.651 |
| | LPIPS↓ | 0.536 | 0.477 | 0.443 | 0.394 | 0.420 | 0.414 | 0.385 | **0.402** | 0.387 | 0.397 | 0.386 | 0.377 | 0.400 | 0.412 | 0.408 | 0.389 | 0.383 | 0.378 | 0.378 | 0.388 | 0.359 | 0.388 |
| | MANIQA↑ | 0.354 | 0.335 | 0.346 | 0.307 | 0.301 | 0.308 | 0.314 | **0.342** | 0.289 | 0.310 | 0.314 | 0.294 | 0.323 | 0.291 | 0.299 | 0.326 | 0.314 | 0.301 | 0.309 | 0.294 | 0.303 | 0.305 |
| | MUSIQ↑ | 62.196 | 58.947 | 60.355 | 55.469 | 54.470 | 54.725 | 57.003 | **59.414** | 51.707 | 54.426 | 56.515 | 53.501 | 56.890 | 52.949 | 54.436 | 57.691 | 56.679 | 54.951 | 56.229 | 53.996 | 54.644 | 54.708 |
| | CLIPIQ↑ | 0.555 | 0.530 | 0.557 | 0.468 | 0.467 | 0.487 | 0.495 | **0.518** | 0.460 | 0.493 | 0.513 | 0.464 | 0.517 | 0.461 | 0.484 | 0.516 | 0.511 | 0.492 | 0.496 | 0.487 | 0.464 | 0.482 |
| RealSR | PSNR↑ | 20.309 | 21.933 | 22.268 | 22.971 | 23.288 | 23.892 | 23.987 | **23.718** | 24.156 | 24.413 | 24.038 | 24.069 | 23.754 | 24.223 | 24.472 | 24.282 | 24.541 | 24.659 | 24.726 | 24.668 | 24.660 | 25.079 |
| | SSIM↑ | 0.572 | 0.618 | 0.627 | 0.645 | 0.649 | 0.667 | 0.675 | **0.671** | 0.681 | 0.679 | 0.675 | 0.677 | 0.663 | 0.686 | 0.681 | 0.683 | 0.691 | 0.695 | 0.695 | 0.689 | 0.700 | 0.700 |
| | LPIPS↓ | 0.463 | 0.401 | 0.376 | 0.341 | 0.346 | 0.327 | 0.309 | **0.326** | 0.324 | 0.333 | 0.319 | 0.328 | 0.327 | 0.331 | 0.324 | 0.301 | 0.299 | 0.306 | 0.302 | 0.306 | 0.286 | 0.302 |
| | MANIQA↑ | 0.390 | 0.385 | 0.392 | 0.342 | 0.323 | 0.329 | 0.344 | **0.355** | 0.305 | 0.297 | 0.326 | 0.320 | 0.342 | 0.306 | 0.304 | 0.324 | 0.331 | 0.302 | 0.313 | 0.311 | 0.318 | 0.294 |
| | MUSIQ↑ | 62.915 | 61.800 | 62.705 | 57.016 | 56.058 | 57.561 | 59.083 | **60.231** | 53.527 | 51.093 | 55.810 | 55.300 | 57.577 | 53.289 | 53.097 | 55.962 | 55.995 | 52.918 | 53.569 | 53.755 | 54.880 | 51.643 |
| | CLIPIQ↑ | 0.539 | 0.555 | 0.573 | 0.495 | 0.480 | 0.498 | 0.503 | **0.525** | 0.451 | 0.445 | 0.500 | 0.478 | 0.505 | 0.434 | 0.459 | 0.488 | 0.484 | 0.445 | 0.441 | 0.460 | 0.457 | 0.428 |
| DrealSR | PSNR↑ | 22.194 | 23.889 | 24.366 | 24.637 | 24.615 | 25.348 | 25.509 | **25.294** | 25.750 | 26.239 | 25.738 | 25.735 | 25.296 | 25.656 | 25.854 | 25.462 | 25.610 | 25.928 | 25.935 | 25.892 | 26.077 | 26.087 |
| | SSIM↑ | 0.623 | 0.653 | 0.658 | 0.667 | 0.664 | 0.679 | 0.691 | **0.691** | 0.700 | 0.704 | 0.695 | 0.706 | 0.676 | 0.699 | 0.692 | 0.687 | 0.690 | 0.704 | 0.704 | 0.705 | 0.711 | 0.707 |
| | LPIPS↓ | 0.503 | 0.446 | 0.410 | 0.378 | 0.396 | 0.383 | 0.360 | **0.374** | 0.355 | 0.371 | 0.367 | 0.358 | 0.378 | 0.377 | 0.377 | 0.358 | 0.360 | 0.365 | 0.351 | 0.363 | 0.334 | 0.355 |
| | MANIQA↑ | 0.393 | 0.388 | 0.378 | 0.361 | 0.328 | 0.328 | 0.336 | **0.363** | 0.299 | 0.297 | 0.323 | 0.305 | 0.339 | 0.279 | 0.298 | 0.331 | 0.316 | 0.298 | 0.318 | 0.297 | 0.311 | 0.295 |
| | MUSIQ↑ | 63.069 | 61.294 | 60.912 | 60.512 | 56.455 | 56.684 | 58.631 | **59.925** | 53.468 | 51.483 | 56.183 | 53.871 | 57.318 | 50.745 | 53.078 | 57.054 | 54.372 | 53.331 | 55.631 | 52.445 | 54.312 | 51.880 |
| | CLIPIQ↑ | 0.596 | 0.615 | 0.616 | 0.583 | 0.538 | 0.534 | 0.552 | **0.572** | 0.497 | 0.488 | 0.541 | 0.499 | 0.545 | 0.451 | 0.508 | 0.560 | 0.523 | 0.497 | 0.522 | 0.489 | 0.498 | 0.473 |
| RealLQ250 | MANIQA↑ | 0.360 | 0.339 | 0.344 | 0.316 | 0.307 | 0.319 | 0.316 | **0.353** | 0.303 | 0.319 | 0.321 | 0.311 | 0.336 | 0.313 | 0.322 | 0.325 | 0.329 | 0.312 | 0.322 | 0.313 | 0.307 | 0.320 |
| | MUSIQ↑ | 66.190 | 62.542 | 62.444 | 58.197 | 58.569 | 59.291 | 60.151 | **63.867** | 55.879 | 57.670 | 59.747 | 56.909 | 61.776 | 59.013 | 59.334 | 60.124 | 59.849 | 59.368 | 59.506 | 60.404 | 58.461 | 60.373 |
| | CLIPIQ↑ | 0.583 | 0.552 | 0.575 | 0.505 | 0.505 | 0.522 | 0.504 | **0.557** | 0.497 | 0.533 | 0.533 | 0.503 | 0.543 | 0.509 | 0.528 | 0.543 | 0.534 | 0.514 | 0.519 | 0.527 | 0.497 | 0.527 |

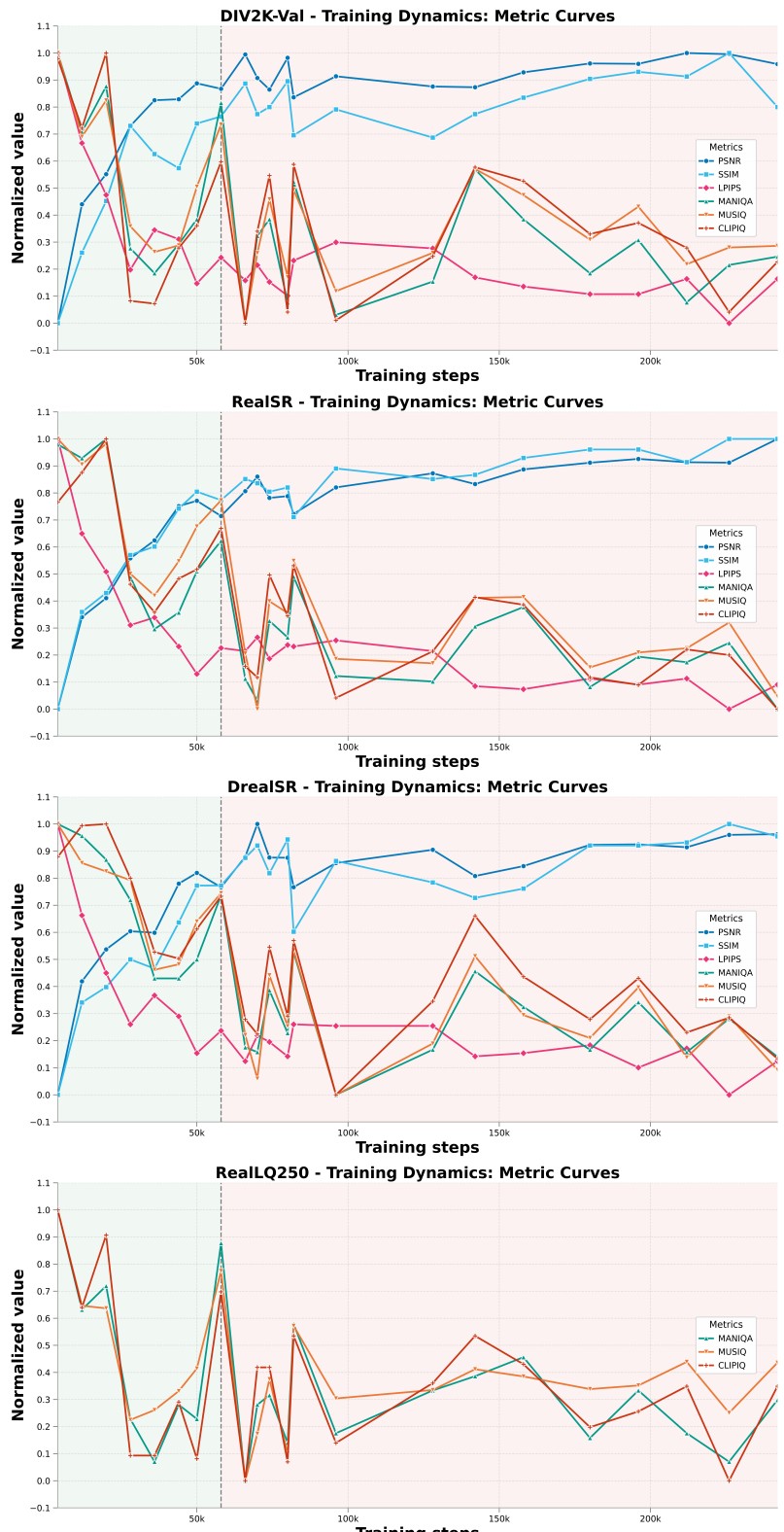

Figure 8: **Visualization of the Knee-Point Phenomenon.** The curves illustrate the evolution of 6metrics throughout the training process. A distinct "Knee-Point" is observable around step 58k, marked by a peak in performance and stability, followed by an oscillation phase where metrics fluctuate or degrade. This visual evidence corroborates the quantitative data in Table 8.

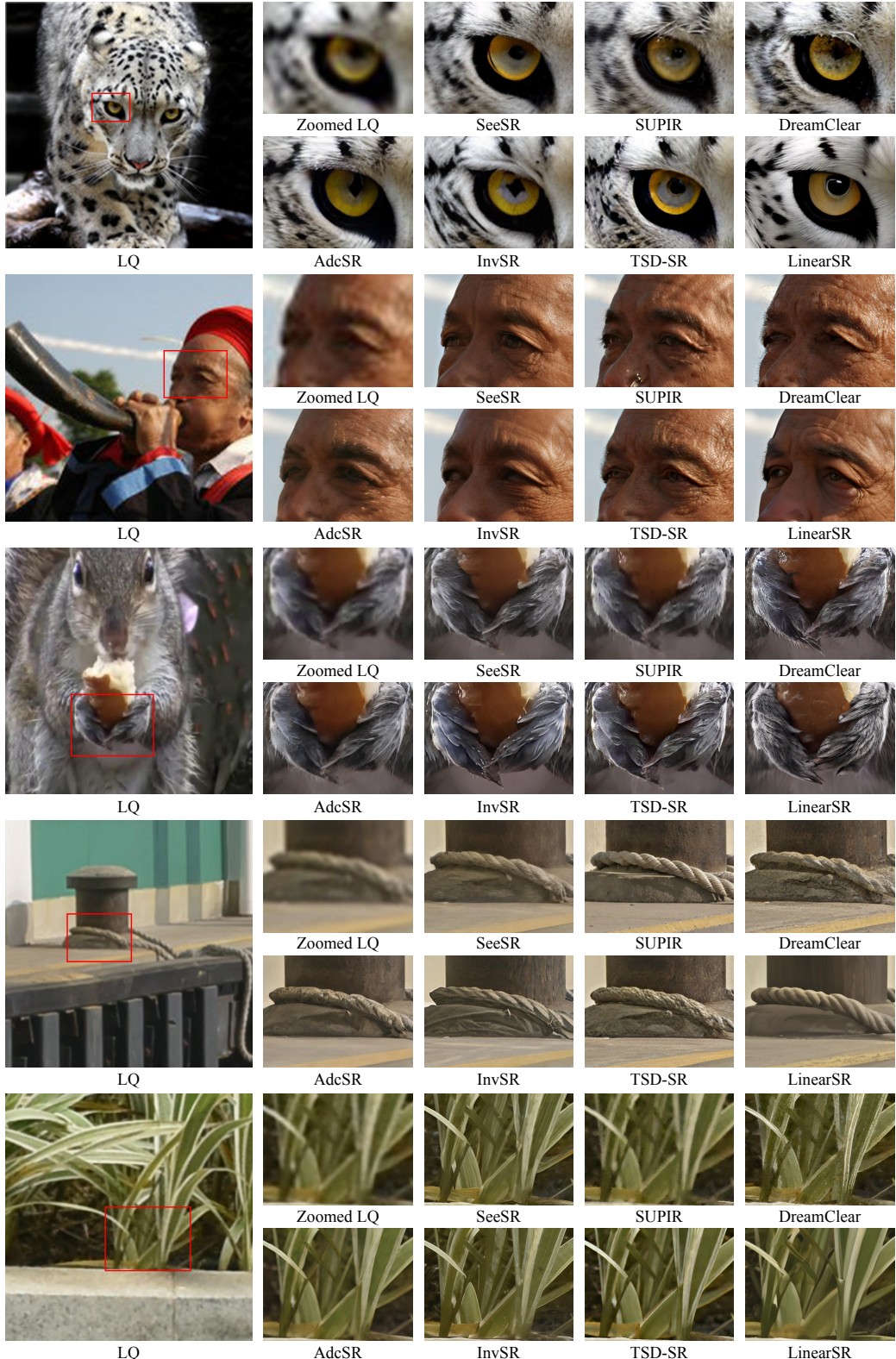

Figure 9: **Visual comparisons on real-world degraded images.** Our **LinearSR** consistently produces more realistic details and textures while maintaining high fidelity.

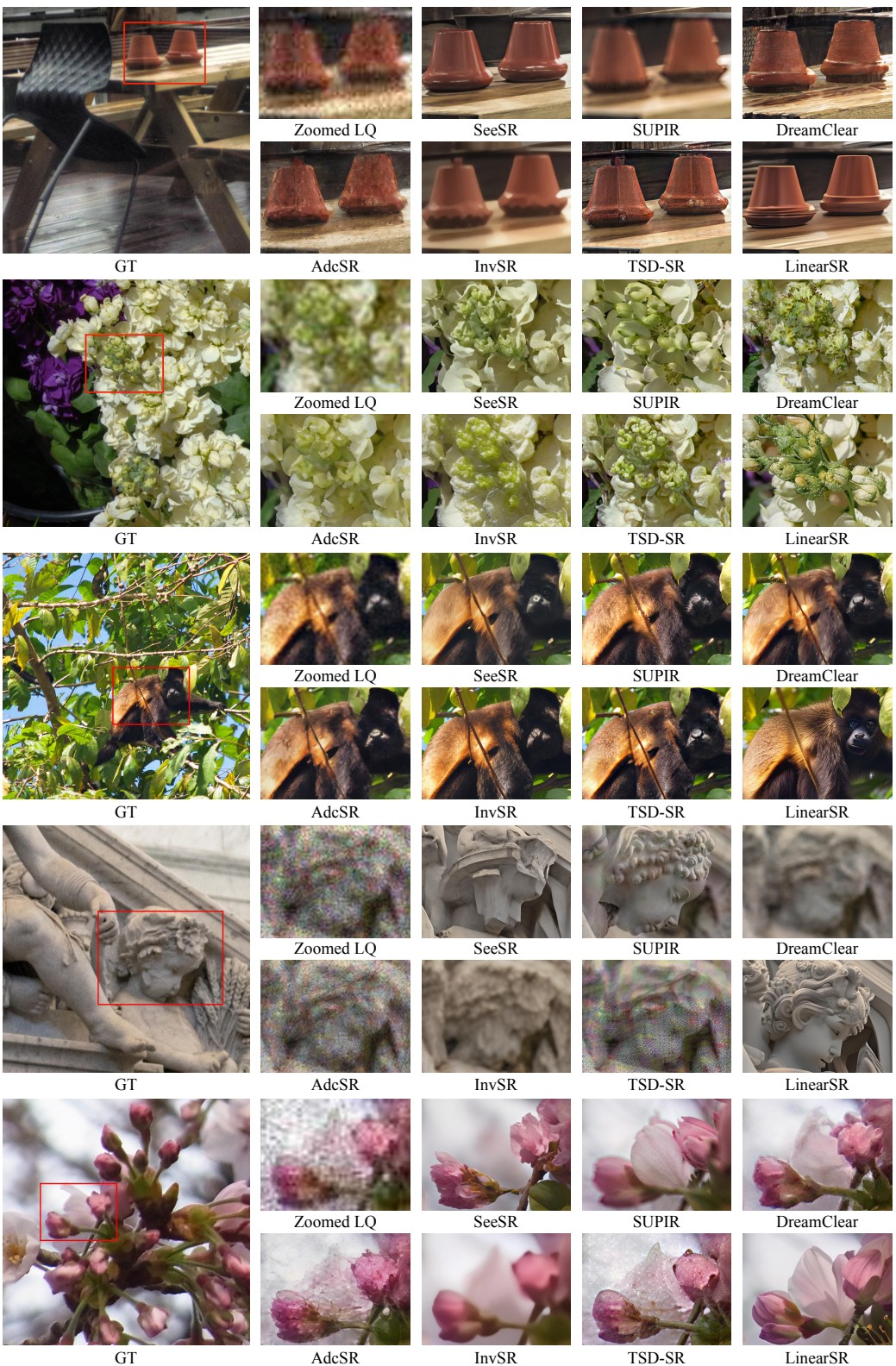

Figure 10: **Visual comparisons on synthetically degraded images.** Compared against the GT, **LinearSR** demonstrates superior performance in restoring faithful details under severe degradation.

