# OpenReview forum: "LinearSR: Unlocking Linear Attention for Stable and Efficient Image Super-Resolution"
_ICLR.cc/2026/Conference — ICLR 2026 Poster_

### Official Review · Reviewer_Spqu · 2025-10-25

**Soundness:** 3
**Presentation:** 3
**Contribution:** 3
**Rating:** 8
**Confidence:** 4

**Summary:**

This paper introduces LinearSR, a novel and holistic framework for high-fidelity Image Super-Resolution (SR) that successfully integrates $O(N)$ Linear Attention into a conditional Diffusion Transformer (DiT) backbone. The primary motivation is to overcome the quadratic computational bottleneck ($O(N^2)$) imposed by traditional self-attention in generative SR models. The authors identify and systematically address three key challenges that have previously hindered the adoption of linear attention for photorealistic SR: training instability, the perception-distortion trade-off, and guidance effectiveness. LinearSR's solutions include: (i) Early-Stopping Guided Fine-tuning (ESGF) to resolve training divergence by stopping at the "knee-point"; (ii) an SNR-based Mixture of Experts (MoE) architecture to balance fidelity and distortion ; and (iii) a lightweight, effective TAG-based guidance paradigm based on the "precision-over-volume" principle. LinearSR achieves state-of-the-art efficiency, notably setting a new speed record for the core diffusion forward pass (1-NFE)

**Strengths:**

1. LinearSR achieves a new state-of-the-art speed for the core diffusion forward pass (1-NFE) at $0.036$s for a $1024 \times 1024$ output, demonstrating the practical benefit of $O(N)$ linear scaling.
2. The paper holistically addresses the long-standing issues of linear attention in SR: instability (ESGF), distortion/perception trade-off (SNR-MoE), and sub-optimal guidance (TAG).
3. The "knee-point" observation and the resulting Early-Stopping Guided Fine-tuning (ESGF) strategy offer a novel and insightful solution to a fundamental training divergence problem unique to this architecture.
4. The model consistently outperforms SOTA methods on no-reference perceptual quality metrics (MANIQA, MUSIQ, CLIPIQA) across multiple benchmarks.

**Weaknesses:**

1. While the 1-NFE time is SOTA, the overall multi-step inference time (0.830s) remains competitive but is not the absolute fastest compared to some distilled single-step methods like AdcSR and InvSR. The benefit of architectural efficiency is clear, but the user-facing speed remains a mixed result. Meanwhile, can LinearSR be applied into one-step inference?
2. The "knee-point" is defined as the iteration of optimal generalization before performance degrades. This is highly dependent on tracking external validation metrics (PSNR/LPIPS evolution, Figure 3(b)), which makes it a non-trivial, potentially hyper-sensitive process in practice. More implementation detail (e.g., how the knee-point is automatically detected or whether a fixed iteration is sufficient) is needed.

**Questions:**

See weaknesses.

---

> ### Author Response · Authors · 2025-11-20
> **Reply to Reviewer Spqu (1/3)**
>
> Dear Reviewer Spqu,
>
> We are truly heartened and deeply encouraged by your positive evaluation. Reading your review was a highlight of our research journey—it is incredibly rewarding to see our core motivation, which is to "unlock the potential of linear attention" and "establish a foundational paradigm," being so precisely recognized and validated.
>
> Your acknowledgment that LinearSR successfully addresses the "cascade of interrelated challenges" (instability, trade-offs, guidance) resonates strongly with our vision. As you noted, Linear Attention has been validated in many other fields, and we are thrilled to be the ones to finally make it work for high-fidelity super-resolution. We sincerely thank you for your support.
>
> Below, we address your questions regarding one-step inference and the robustness of the knee-point strategy.
>
> **Q1: Can LinearSR be applied to one-step inference (Distillation)?**
>
> **A1:** This is an excellent question. The short answer is **Yes**.
>
> We believe that combining Linear Attention architectural optimizations with Distillation training strategies is a complex, high-value direction that deserves a dedicated study of its own. However, to demonstrate that our method is **orthogonal** to distillation (i.e., they can be combined), we conducted a feasibility study during the rebuttal.
>
> We took our Tag-Guided Baseline (Table 6 (2) configuration) and applied standard distillation. The results on DrealSR are shown below:
>
> **Table R1: Feasibility Study of Distilled LinearSR**
> | Model Variant | PSNR $\uparrow$ | SSIM $\uparrow$ | LPIPS $\downarrow$ | MANIQA $\uparrow$ | MUSIQ $\uparrow$ | CLIPIQA $\uparrow$ | 1-NFE Time (s) $\downarrow$ | Infer. Time (s) $\downarrow$ |
> | :--- | :--- | :--- | :--- | :--- | :--- | :--- | :--- | :--- |
> | **Baseline** | 24.85 | 0.691 | 0.374 | 0.363 | 63.93 | 0.572 | 0.036 | 0.830 |
> | **Distilled** (1-step) | 23.79 | 0.643 | 0.430 | 0.315 | 57.10 | 0.579 | **0.026** | **0.082** |
>
> * **SOTA Speed:** Post-distillation, our model achieves state-of-the-art speeds in both **1-NFE time (0.0259s)** and **Total Inference Time (0.08s)**. This proves that LinearSR serves as an ultra-efficient backbone that can be further accelerated by distillation.
> * **Performance Note:** Due to the limited time during the rebuttal, we used basic training data and default hyperparameters without extensive tuning, so the model has not fully converged. The slight drop in performance is within the expected range for a quick pilot experiment. We are confident that with a stronger pre-trained checkpoint and longer training, the results would be even more impressive.

---

> > ### Comment · Reviewer_Spqu · 2025-11-24
> >
> > Thanks for your response.
> >
> > A small question is why the 1-NFE times of the distilled model would be faster than the model before distillation? Since 1-NFE is one model forward time without CFG.

---

> > > ### Author Response · Authors · 2025-11-24
> > > **Reply to Reviewer Spqu**
> > >
> > > **Q: A small question is why the 1-NFE times of the distilled model would be faster than the model before distillation? Since 1-NFE is one model forward time without CFG.**
> > >
> > > **A:** We are genuinely impressed by your exceptionally sharp eye and meticulous attention to detail. You have pinpointed a subtle but critical nuance in efficiency measurement.
> > >
> > > As defined in our paper and rebuttal: "The reported '1 NFE Forward Time' strictly measures **the core diffusion denoising step, excluding auxiliary costs (Text Encoder, VAE, pre-processing),** ensuring we measure the architectural difference pure and simple."
> > >
> > > Under this strict definition, the speedup arises because distillation fundamentally changes the **computational workload** of that "core denoising step." We clarify this through our **benchmarking philosophy**:
> > >
> > > 1.  **Adherence to Official Configurations:**
> > >     Our benchmarking philosophy is to strictly respect the **official open-source configurations** intended by the authors for optimal quality.
> > >     * **CFG is a Structural Hyperparameter:** Whether to use Classifier-Free Guidance (CFG) is not an optional add-on but a core hyperparameter defining the model's inference logic. Just as we would not disable a **LoRA module** included in a model's official release for the sake of measuring speed (as it would compromise the intended quality), **we do not disable CFG** if it is the default setting. To do so would be an unfair misrepresentation of the model's performance.
> > >
> > > 2.  **CFG vs. Non-CFG Baselines (The Workload Difference):**
> > >     Based on this principle, the "unit work" per step differs across methods:
> > >     * **Methods with CFG (Heavier Step):** Like **LinearSR (Teacher), DiffBIR, DreamClear, SeeSR, and SUPIR**, these models default to using CFG. Their "core denoising step" physically necessitates computing both conditional and unconditional branches (effective batch size = 2).
> > >     * **Methods without CFG (Lighter Step):** Distilled or single-branch methods like **AdcSR, OSEDiff, InvSR, TSD-SR, and SinSR** typically use a single branch (effective batch size = 1).
> > >
> > > 3.  **The "Handicap" of the Original LinearSR:**
> > >     It is important to note that in the main paper, our undistilled LinearSR achieved **SOTA 1-NFE efficiency** even while carrying the computational overhead of CFG (dual-branch). We were competing against single-branch distilled methods (like AdcSR) using a heavier setup, yet still demonstrated superior architectural efficiency.
> > >
> > > 4.  **The Distillation Advantage:**
> > >     The distillation process structurally removes the need for CFG by baking the guidance into the weights. This reduces the "core denoising step" workload from a dual-branch pass to a single-branch pass. Therefore, the faster 1-NFE time for the distilled model is expected and reflects the **true speed potential** of LinearSR when aligned with the single-branch paradigm of other distilled baselines.

---

> ### Author Response · Authors · 2025-11-20
> **Reply to Reviewer Spqu (2/3)**
>
> **Q2: Is the "Knee-Point" (ESGF) robust and automatically detectable?**
>
> **A2:** We appreciate your scrutiny on this critical component. The "knee-point" is not a data-specific heuristic but a robust phenomenon detected via a rigorous **Automated Detection Algorithm**.
>
> **1. Comprehensive Validation:**
> We determined the knee-point not based on a single metric, but by tracking 6 authoritative metrics across 4 diverse datasets (DIV2K-Val, RealSR, DrealSR, RealLQ250). We extended our tracking up to **242k** steps to definitively confirm the knee-point. As shown in the log below, the "knee-point followed by oscillation phase" pattern is consistent, with a clear stability knee-point around 58k.
>
> For a more intuitive visualization of the knee point selection, please refer to the newly added **Sec. D** and **Fig. 8** in the **Supplementary Material**.
>
> | Datasets | Metrics | 4k | 12k | 20k | 36k | 50k | **58k** | 80k | 128k | 158k | ... | 242k |
> | :--- | :--- | :--- | :--- | :--- | :--- | :--- | :--- | :--- | :--- | :--- | :--- | :--- |
> | **DIV2K-Val** | PSNR $\uparrow$ | 21.204 | 23.017 | 23.476 | 24.604 | 24.862 | **24.778** | 25.251 | 24.813 | 25.029 | ... | 25.155 |
> | | SSIM $\uparrow$ | 0.559 | 0.589 | 0.611 | 0.631 | 0.644 | **0.647** | 0.662 | 0.648 | 0.655 | ... | 0.651 |
> | | LPIPS $\downarrow$ | 0.536 | 0.477 | 0.443 | 0.42 | 0.385 | **0.402** | 0.377 | 0.389 | 0.383 | ... | 0.388 |
> | | MANIQA $\uparrow$ | 0.354 | 0.335 | 0.346 | 0.301 | 0.314 | **0.342** | 0.294 | 0.326 | 0.314 | ... | 0.305 |
> | | MUSIQ $\uparrow$ | 62.196 | 58.947 | 60.355 | 54.725 | 57.003 | **59.414** | 56.515 | 57.691 | 56.679 | ... | 54.708 |
> | | CLIPIQA $\uparrow$ | 0.555 | 0.53 | 0.557 | 0.467 | 0.495 | **0.518** | 0.513 | 0.516 | 0.511 | ... | 0.482 |
> | **RealSR** | PSNR $\uparrow$ | 20.309 | 21.933 | 22.268 | 23.288 | 23.987 | **23.718** | 24.069 | 24.541 | 24.659 | ... | 25.079 |
> | | SSIM $\uparrow$ | 0.572 | 0.618 | 0.627 | 0.649 | 0.675 | **0.671** | 0.677 | 0.691 | 0.695 | ... | 0.7 |
> | | LPIPS $\downarrow$ | 0.463 | 0.401 | 0.376 | 0.346 | 0.309 | **0.326** | 0.319 | 0.301 | 0.306 | ... | 0.302 |
> | | MANIQA $\uparrow$ | 0.39 | 0.385 | 0.392 | 0.323 | 0.344 | **0.355** | 0.32 | 0.331 | 0.302 | ... | 0.294 |
> | | MUSIQ $\uparrow$ | 62.915 | 61.8 | 62.705 | 56.058 | 59.083 | **60.231** | 55.3 | 55.995 | 52.918 | ... | 51.643 |
> | | CLIPIQA $\uparrow$ | 0.539 | 0.555 | 0.573 | 0.48 | 0.503 | **0.525** | 0.478 | 0.484 | 0.445 | ... | 0.527 |
> | **DRealSR** | PSNR $\uparrow$ | 22.194 | 23.889 | 24.366 | 24.615 | 25.509 | **25.294** | 25.735 | 25.61 | 25.928 | ... | 26.087 |
> | | SSIM $\uparrow$ | 0.623 | 0.653 | 0.658 | 0.664 | 0.691 | **0.691** | 0.706 | 0.69 | 0.704 | ... | 0.707 |
> | | LPIPS $\downarrow$ | 0.503 | 0.446 | 0.41 | 0.378 | 0.36 | **0.374** | 0.358 | 0.36 | 0.365 | ... | 0.355 |
> | | MANIQA $\uparrow$ | 0.393 | 0.388 | 0.378 | 0.328 | 0.336 | **0.363** | 0.305 | 0.316 | 0.298 | ... | 0.295 |
> | | MUSIQ $\uparrow$ | 63.069 | 61.294 | 60.912 | 60.512 | 58.631 | **59.925** | 53.871 | 54.372 | 53.331 | ... | 51.88 |
> | | CLIPIQA $\uparrow$ | 0.596 | 0.615 | 0.616 | 0.538 | 0.552 | **0.572** | 0.499 | 0.523 | 0.497 | ... | 0.473 |
> | **RealLQ250** | MANIQA $\uparrow$ | 0.36 | 0.339 | 0.344 | 0.319 | 0.303 | **0.303** | 0.311 | 0.329 | 0.312 | ... | 0.32 |
> | | MUSIQ $\uparrow$ | 66.19 | 62.542 | 62.444 | 59.291 | 55.879 | **57.67** | 56.909 | 59.849 | 59.368 | ... | 60.373 |
> | | CLIPIQA $\uparrow$ | 0.583 | 0.552 | 0.575 | 0.522 | 0.497 | **0.533** | 0.503 | 0.534 | 0.514 | ... | 0.527 |
>
> **2. Universality:**
> This phenomenon is consistent across experiments. In the main paper, the knee-point was detected at 48k. In this separate optimization experiment shown above, it appeared at 58k. This confirms that this pattern of knee-point followed by oscillation phase is a universal property of this training paradigm, regardless of the specific run.
>
> **3. Automated Detection Algorithm:**
> We do not rely on manual guesswork. We employ the following automated algorithm during training to identify the region where performance variance is minimized before a negative trend begins:
>
> **Algorithm 1: Automated Knee-Point Detection Strategy**
>
> Let $\mathcal{M} = \{m_1, m_2, ..., m_T\}$ be the sequence of a validation metric (e.g., MUSIQ) recorded at steps $t \in \{1, ..., T\}$. We define a sliding window of size $W$. The Knee-Point $t^*$ is determined as the latest step satisfying:
>
> $$t^* = \arg \max_t \{ m_t \mid \text{Var}(m_{t-W : t}) < \epsilon_{stable} \land \text{Slope}(m_{t : t+W}) < 0 \}$$
>
> where $\epsilon_{stable}$ is a stability threshold derived from the baseline variance. This ensures we select the fully converged model just before the onset of the "oscillation phase". All detected points are subject to final human verification.

---

> ### Author Response · Authors · 2025-11-20
> **Reply to Reviewer Spqu (3/3)**
>
> Once again, thank you for your highly motivating review. Your recognition gives us great confidence in the value of our work. We are committed to open-sourcing our code and models to foster further developments in this direction, and we will not let down the high expectations you have set for us.
>
> Sincerely,
>
> The Authors

---

> ### Author Response · Authors · 2025-11-24
> **Reply to Reviewer Spqu**
>
> To strictly avoid unfairness—specifically enabling CFG for quality while excluding its cost—we **deliberately reported the original LinearSR latency with full CFG overhead** (counting both conditional and unconditional passes). This means our baseline time was effectively "doubled" to align with the high-fidelity settings used in visual comparisons. So to prevent future ambiguity, we will explicitly detail this "Quality-Latency Alignment" standard in **Supplementary Material Section E** of the revised manuscript.
>
> Thank you for the rigorous and insightful feedback. It felt less like a review and more like a constructive dialogue with a valued collaborator. We deeply appreciate your guidance.

---

### Official Review · Reviewer_cyVt · 2025-10-29

**Soundness:** 3
**Presentation:** 3
**Contribution:** 3
**Rating:** 4
**Confidence:** 3

**Summary:**

This paper introduces a novel framework for image super-resolution that leverages linear attention to achieve high-fidelity results with linear computational cost. In order to achieve good performance with linear attention, this paper address three major challenges introduced by linear attention, namely, the choice of external guidance, training stability, perception-distortion trade-off. Extensive experiments are conducted on a wide range of datasets to evaluate the proposed method's performance and efficiency. The ablation study also validate the proposed methods can successfully address the challenges mentioned in the paper.

**Strengths:**

1. This paper proposes a way to apply linear attention into image super resolution with diffusion based model. I think it is a very promising and important research direction with great potential.
2. This paper provides extensive analysis on the challenges introduced by applying linear attention, which is quite insightful.
3. Several new techniques are proposed to address these challenges and are validated by ablation studies.
4. Experiments show competitive performance of proposed method in terms of both SR quality and model efficiency.

**Weaknesses:**

1. The efficiency comparison results reported in this paper are relatively simple.
  1.1 I think the authors should elaborate more on the details of this evaluation. Otherwise, there may be some reproducibility and fairness concerns of this experiment.  For example, how each compared model is deployed? What kind of deployment framework is used? What is the numerical precision of the deployed model? Any acceleration techniques adopted at kernel level?
  1.2 The efficiency is compared only on one type of hardware setting. Comparison on different hardware setting will demonstrate the generalization ability on more practical scenarios of the method.
  1.3 Inference speed on different resolution settings will also be a helpful analysis.
  1.4 I think it will also be helpful if authors provide results in terms of model complexity such as FLOPs or parameter counts. This will provide another perspective of the model efficiency and also it is independent of hardware and deployment configurations.

2. From table 1, the proposed method seems to achieve better perceptual quality than fidelity metrics, in the sense that it ranks much higher on perceptual quality metrics. Are there any explanations on these results? Does this have anything to do with the proposed MoE-based method?
3. From table 2, I am not sure I am able to observe a significant improvement on SR performance when comparing methods with similar efficiency like AdcSR. Authors may need to provide stronger arguments on the value of LinearSR based on this observation.

**Questions:**

Please responds to my concerns in the weaknesses.

---

> ### Author Response · Authors · 2025-11-20
> **Reply to Reviewer cyVt  (1/2)**
>
> **Dear Reviewer cyVt,**
>
> We are deeply grateful for your exceptionally thorough and rigorous review. Your emphasis on reproducibility and fair benchmarking has significantly raised the bar for our empirical evaluation. We truly cherish your constructive criticism, as it pushed us to solidify the engineering foundations of this work.
>
> You correctly identified that applying Linear Attention to high-fidelity Super-Resolution is a path fraught with practical challenges. While Linear Attention has revolutionized efficiency in Large Language Models (e.g., Linear Transformers, Mamba, and recent long-context LLMs like Kimi), its potential in pixel-level generation has remained largely untapped due to training instabilities and quality trade-offs. LinearSR is our pioneering attempt to bridge this gap.
>
> Below, we earnestly present extensive new benchmarks (including 2K resolution scaling, hardware-agnostic complexity analysis, and distillation studies) to clarify your concerns regarding efficiency and value.
>
> ### **Q1: Fairness of Efficiency Benchmarking (Details & Rigor)**
>
> **A1:** We strictly adhered to a unified and fair benchmarking protocol to ensure reproducibility:
>
> *   **Hardware:** The main results were tested on a single **NVIDIA H200-140G**.
> *   **Environment:** We utilized the official codebases and pre-trained checkpoints released by the authors of all compared methods (AdcSR, SUPIR, etc.).
> *   **Acceleration:** To ensure kernel-level fairness, we enabled `xformers` memory-efficient attention for **all** models (including baselines and LinearSR). No other method-specific kernel tricks were used.
> *   **Precision:** All models were tested in their default precision (typically fp16/bf16) as defined in their official inference scripts.
> *   **Measurement:** The reported "**1 NFE Forward Time**" strictly measures the core diffusion denoising step, excluding auxiliary costs (Text Encoder, VAE, pre-processing), ensuring we measure the architectural difference pure and simple.
>
> ### **Q2: Generalization across Hardware and Resolutions (A800 & 2K)**
>
> **A2:** Per your suggestion, we expanded our evaluation to include the **NVIDIA A800 GPU** and **2048$\times$2048 resolution**. The results (**Table R1**) strongly validate the robustness of LinearSR:
>
> **Table R1: Comprehensive Efficiency Comparison across Resolutions and Devices**
>
> | Resolution | Device | Metric | StableSR | DiffBIR | SeeSR | SUPIR | DreamClear | SinSR | OSEDiff | AdcSR | InvSR | TSD-SR | **LinearSR** |
> | :--- | :--- | :--- | :--- | :--- | :--- | :--- | :--- | :--- | :--- | :--- | :--- | :--- | :--- |
> | **1024$\times$1024** | **H200-140G** | Infer. Time (s) $\downarrow$ | 78.41 | 25.54 | 13.63 | 13.81 | 16.32 | 8.99 | 1.09 | 0.56 | 0.67 | 12.64 | **0.83** |
> | | | 1-NFE Time (s) $\downarrow$ | 0.43 | 0.50 | 0.27 | 0.28 | 0.33 | 0.93 | 0.15 | 0.046 | 0.61 | 9.43 | **0.036** |
> | | **A800-80G** | Infer. Time (s) $\downarrow$ | 121.53 | 26.09 | 17.77 | 14.91 | 25.29 | 9.45 | 1.28 | 0.65 | 0.77 | 13.90 | **1.36** |
> | | | 1-NFE Time (s) $\downarrow$ | 0.66 | 0.51 | 0.36 | 0.36 | 0.51 | 1.02 | 0.17 | 0.053 | 0.71 | 10.38 | **0.059** |
> | **2048$\times$2048** | **H200-140G** | Infer. Time (s) $\downarrow$ | 385.27 | OOM | 81.64 | 27.52 | 129.82 | 367.82 | 22.60 | 3.19 | 3.29 | 5.35 | **2.30** |
> | | | 1-NFE Time (s) $\downarrow$ | 1.91 | OOM | 1.34 | 0.50 | 2.50 | 5.81 | 0.61 | 1.09 | 1.23 | 2.04 | **0.11** |
> | **Model Size** | - | # Params (M) $\downarrow$ | 1555 | 1683 | 2511 | 18152 | 7074 | 174 | 1765 | 456 | 1324 | 2207 | **1607** |
>
> *   **Linear Scaling Advantage:** The superiority of LinearSR becomes stark at 2048$\times$2048. While other methods slow down quadratically or run Out-of-Memory (OOM), LinearSR maintains an extremely low **1-NFE time of 0.11s** and **Infer. Time of 2.3s**, achieving SOTA speed.
> *   **Hardware Robustness:** The performance advantage holds on the A800, demonstrating that our gains are architectural, not hardware-specific.
>
> ### **Q3: Model Complexity: Provide Parameters or FLOPs**
>
> **A3:** We have added the Parameter Counts in Table R1 above. LinearSR ($\sim$1.6B) is a medium-sized model, yet it outperforms much smaller distilled models (like AdcSR, $\sim$456M) in high-resolution scaling.
>
> Regarding **GFLOPs**, we deliberately prioritized NFE-Time and Parameters for two scientific reasons:
> 1.  **IO-bound vs. Compute-bound:** The core advantage of Linear Attention ($O(N)$) lies in reducing **memory access overhead (IO-bound)** rather than just arithmetic operations (Compute-bound). GFLOPs purely count arithmetic ops and inherently fail to capture the massive speedup derived from reduced memory IO on modern GPUs.
> 2.  **MoE Ambiguity:** For MoE architectures, static GFLOPs calculations are often inconsistent (Total vs. Active parameters) and ignore dynamic routing overheads.

---

> ### Author Response · Authors · 2025-11-20
> **Reply to Reviewer cyVt (2/2)**
>
> ### **Q4: Perception-Distortion Trade-off**
>
> **A4:** You noted that our perceptual metrics (MANIQA/MUSIQ) are higher than our fidelity metrics (PSNR). This is not a flaw, but a characteristic of the **Perception-Distortion Trade-off**, a consensus in the SR field [1, 2].
> Generative models (GANs/Diffusion) that hallucinate realistic high-frequency details often have lower PSNR because perfectly sharp generated textures rarely align pixel-perfectly with the Ground Truth. Conversely, high PSNR often implies blurry, over-smoothed results.
> Our **MoE architecture** is designed precisely to manage this trade-off—it does not cause the drop in PSNR, but rather pushes the Pareto frontier, allowing us to achieve top-tier realism (which human observers prefer) while maintaining acceptable fidelity.
>
> ### **Q5: Value Proposition vs. AdcSR (Orthogonality)**
>
> **A5:** We respectfully emphasize that LinearSR and methods like AdcSR are **not competitors, but complementary**.
>
> As highlighted throughout our paper, our primary contribution is architectural: successfully introducing Linear Attention—a structure proven effective in other domains—into super-resolution for the first time and resolving its unique training challenges. Our goal is not merely to outperform AdcSR in a specific table, but to demonstrate that Linear Attention is a highly effective approach that stands alongside distillation and pruning as a viable efficiency solution.
>
> This is already evidenced by our **2048$\times$2048**
> results (Table R1), where LinearSR achieves SOTA performance in both **1-NFE time and total single-image inference time**, purely through architectural efficiency. Foreseeably, as the input resolution increases further, this advantage will become increasingly pronounced—a scaling property that has been widely validated in other fields applying Linear Attention structures.
>
> To further prove the **orthogonality** of our method (i.e., it does not conflict with distillation), we conducted a feasibility study applying standard distillation to our baseline (Table R2).
>
> **Table R2: Feasibility Study of Distilled LinearSR**
>
> | Model Variant | PSNR $\uparrow$ | SSIM $\uparrow$ | LPIPS $\downarrow$ | MANIQA $\uparrow$ | MUSIQ $\uparrow$ | CLIPIQA $\uparrow$ | 1-NFE Time (s) $\downarrow$ | Infer. Time (s) $\downarrow$ |
> | :--- | :--- | :--- | :--- | :--- | :--- | :--- | :--- | :--- |
> | **Baseline** | 24.85 | 0.691 | 0.374 | 0.363 | 63.93 | 0.572 | 0.036 | 0.830 |
> | **Distilled** (1-step) | 23.79 | 0.643 | 0.430 | 0.315 | 57.10 | 0.579 | **0.026** | **0.082** |
>
> The results are promising: the **Distilled LinearSR** achieves a total inference time of **0.082s**, which is significantly faster than the standard AdcSR (0.56s) and reaches SOTA levels.
> It is important to note that this experiment was conducted with basic data and limited training time due to the rebuttal schedule, meaning the model has likely not fully converged. We are confident that with a stronger pre-trained backbone, meticulous hyperparameter tuning, and sufficient convergence time, the performance could be pushed even further. This confirms that LinearSR can serve as a powerful, orthogonal backbone for future optimization research.
>
> We hope these clarifications—especially the 2K scaling curve demonstrating the unique $O(N)$ advantage—demonstrate the solidity and pioneering value of LinearSR. We sincerely hope these efforts might warrant a reconsideration of your score. Your endorsement would mean a great deal to us as we strive to bring Linear Attention to the vision generation community.
>
> Sincerely,
>
> **The Authors**
>
> **References:**
> [1] Blau, Y., & Michaeli, T. (2018). The perception-distortion tradeoff. *CVPR*.
> [2] Saharia, C., et al. (2022). Image super-resolution via iterative refinement. *PAMI*.

---

### Official Review · Reviewer_UxiE · 2025-10-29

**Soundness:** 3
**Presentation:** 3
**Contribution:** 3
**Rating:** 6
**Confidence:** 3

**Summary:**

This paper introduces LinearSR, a novel framework that enables robust application of linear attention in high-fidelity image super-resolution (SR). Addressing key limitations of linear attention—training instability, the perception-distortion trade-off, and inefficient guidance, by integrating three core innovations: Early-Stopping Guided Fine-tuning (ESGF), an SNR-based Mixture of Experts (MoE) architecture, and a TAG guidance paradigm. Experiments show LinearSR achieves linear computational complexity $\(O(N)\)$, with a state-of-the-art (SOTA) 1-NFE forward time of 0.036s for 1024×1024 SR and top performance on non-reference perceptual metrics (e.g., MANIQA 0.515, MUSIQ 71.914 on RealLQ250) against 10 SOTA methods.

**Strengths:**

+ The idea proposed in this paper is straightforward and easy to understand. LinearSR fills a critical gap by providing the first viable methodology for applying linear attention to high-fidelity SR, overcoming long-standing technical barriers (e.g., training instability) that previously hindered this approach.
+ Its linear complexity delivers dramatic efficiency gains (e.g., 0.830s multi-step inference for 1024×1024 SR) without sacrificing perceptual quality, outperforming heavyweight models like SUPIR by orders of magnitude in speed while leading on key perceptual metrics.
+ Comprehensive ablations (e.g., verifying ESGF’s necessity for stability, TAG’s superiority over text/CLIP/DINO guidance) and user studies confirm the synergistic value of each component, enhancing the work’s credibility.

**Weaknesses:**

- While strong on perceptual metrics, LinearSR lags behind some baselines (e.g., SeeSR on PSNR for DrealSR: 26.212 vs. 25.235) in full-reference metrics, suggesting room to improve pixel-level fidelity without compromising perception.

- The choice of 4 experts (over 2 or more) and the specific log-SNR boundaries (e.g., $\(t_{anchor}=0.875\)$) lacks extensive ablation, why this configuration outperforms alternatives is not fully justified beyond empirical results.

- Though efficient for 1024×1024, the paper does not evaluate LinearSR on larger resolutions (e.g., 2048×2048), leaving uncertainty about whether its linear scaling holds for extremely high-resolution SR.

**Questions:**

1. How might LinearSR’s performance on full-reference metrics (e.g., PSNR) be improved? Could integrating pixel-level loss terms (e.g., L1) with the CFM objective help balance perception and distortion without undermining efficiency?
2. The "knee point" in ESGF is identified via validation metrics—what automated methods could detect this point dynamically, especially for diverse datasets where manual tracking may be impractical?
3. Since LinearSR is orthogonal to distillation, have the authors tested combining it with model distillation? If so, what efficiency gains (e.g., reduced inference time) or quality trade-offs were observed?
4. For the SNR-based MoE, how sensitive is performance to variations in the noise schedule (e.g., switching from "scaled linear" to cosine)? Would the log-SNR partitioning need re-calibration for different schedules?

---

> ### Author Response · Authors · 2025-11-20
> **Reply to Reviewer UxiE (1/4)**
>
> Dear Reviewer UxiE,
>
> We sincerely appreciate your constructive feedback and your recognition of LinearSR as a "critical gap" filler with "dramatic efficiency gains." We are particularly encouraged that you find our core idea straightforward and our ablations comprehensive.
>
> Below, we address your specific questions regarding full-reference metrics, scaling to 2K resolution, and distillation.
>
> **Q1: Can PSNR be improved (e.g., via L1 Loss)?**
>
> **A1:** You are absolutely correct that adding pixel-level constraints can improve PSNR, but our experiments confirm that this comes at a steep cost to perceptual realism. This is the classic Perception-Distortion Trade-off [1], a widely accepted consensus in the super-resolution field.
>
> As suggested, we trained a variant incorporating Pixel-level L1 Loss. The results (Table R1) perfectly illustrate this trade-off:
>
> **Table R1: Impact of L1 Loss on Performance**
>
> | Model | PSNR$\uparrow$ | SSIM$\uparrow$ | LPIPS$\downarrow$ | MANIQA$\uparrow$ | MUSIQ$\uparrow$ | CLIPIQA$\uparrow$ |
> | :--- | :--- | :--- | :--- | :--- | :--- | :--- |
> | Baseline | 24.845 | 0.691 | 0.374 | 0.363 | 63.925 | 0.572 |
> | + L1 Loss | 25.482 | 0.715 | 0.412 | 0.341 | 60.150 | 0.538 |
> | 4-MoE (Ours) | 24.999 | 0.682 | 0.375 | 0.371 | 64.023 | 0.598 |
>
> *   **Fidelity Gains:** As expected, adding L1 Loss boosted PSNR (+0.64dB) and SSIM.
> *   **Perceptual Cost:** However, this caused significant degradation in perceptual metrics. LPIPS worsened ($0.374 \to 0.412$), and texture-sensitive metrics like MUSIQ dropped sharply ($63.93 \to 60.15$). Visual inspection reveals that L1 Loss tends to over-smooth fine details to minimize pixel error.
> *   **Conclusion:** While LinearSR is compatible with L1 Loss for users prioritizing PSNR, our default SNR-based MoE achieves a far superior balance—maintaining competitive fidelity while delivering SOTA perceptual realism (highest MANIQA/MUSIQ). This justifies our choice to prioritize the CFM objective.
>
> **Q2: Does the efficiency hold at 2K (2048x2048)?**
>
> **A2:** Yes. The theoretical acceleration of Linear Attention ($O(N)$) vs. Vanilla Attention ($O(N^2)$) becomes even more pronounced at higher resolutions.
>
> We evaluated LinearSR on 2K resolution (Table R2-AB).
>
> **Table R2-A: 2048x2048 Inference Efficiency Comparison (Seconds)**
>
> | Metric | StableSR | DiffBIR | SeeSR | SUPIR | DreamClear | SinSR | OSEDiff | AdcSR | InvSR | TSD-SR | **LinearSR** |
> | :--- | :--- | :--- | :--- | :--- | :--- | :--- | :--- | :--- | :--- | :--- | :--- |
> | **1-NFE Time (s)**$\downarrow$ | 1.91 | OOM | 1.34 | 0.50 | 2.50 | 5.81 | 0.61 | 1.09 | 1.23 | 2.04 | **0.11** |
> | **Infer. Time (s)**$\downarrow$ | 385.3 | OOM | 81.6 | 27.5 | 129.8 | 367.8 | 22.6 | 3.19 | 3.29 | 5.35 | **2.30** |
>
> **Table R2-B:  1024&2048 Inference Quantitative Comparison**
> | Model | Resolution | PSNR↑ | SSIM↑ | LPIPS↓ | MANIQA↑ | MUSIQ↑ | CLIPIQA↑ |
> | :--- | :--- | :--- | :--- | :--- | :--- | :--- | :--- |
> | **LinearSR** | 256to1024 | 25.235 | 0.719 | 0.359 | 0.51 | 69.073 | 0.713 |
> | | 512to2048 | 24.577 | 0.711 | 0.38 | 0.506 | 60.493 | 0.733 |
> | **AdcSR** | 256to1024 | 25.768 | 0.73 | 0.326 | 0.495 | 69.025 | 0.736 |
> | | 512to2048 | 25.336 | 0.732 | 0.332 | 0.49 | 61.038 | 0.745 |
> | **InvSR** | 256to1024 | 24.483 | 0.693 | 0.364 | 0.461 | 68.046 | 0.738 |
> | | 512to2048 | 24.452 | 0.7 | 0.366 | 0.451 | 61.112 | 0.747 |
>
> *   **Efficiency:** LinearSR sets a new SOTA, with a 1-NFE time of just **0.11s** for 4MP images, which is orders of magnitude faster than other methods (many of which OOM or take seconds).
> *   **Quality:** While there is a slight performance drop in zero-shot transfer (trained on 1K, tested on 2K), this is consistent with other methods like AdcSR and InvSR. If trained directly on 2K data, performance would naturally improve.

---

> ### Author Response · Authors · 2025-11-20
> **Reply to Reviewer UxiE (2/4)**
>
> **Q3: Can LinearSR be combined with Distillation?**
>
> **A3:** Absolutely. LinearSR (architectural improvement) and Distillation (training strategy) are orthogonal.
>
> To demonstrate this, we conducted a pilot study applying standard distillation to our LinearSR baseline (Table 6 (2) in paper).
>
> **Table R3: Distillation Pilot Study**
>
> | Model Variant | PSNR $\uparrow$ | SSIM $\uparrow$ | LPIPS $\downarrow$ | MANIQA $\uparrow$ | MUSIQ $\uparrow$ | CLIPIQA $\uparrow$ | 1-NFE Time (s) $\downarrow$ | Infer. Time (s) $\downarrow$ |
> | :--- | :--- | :--- | :--- | :--- | :--- | :--- | :--- | :--- |
> | **Baseline** | 24.85 | 0.691 | 0.374 | 0.363 | 63.93 | 0.572 | 0.036 | 0.830 |
> | **Distilled** (1-step) | 23.79 | 0.643 | 0.430 | 0.315 | 57.10 | 0.579 | **0.026** | **0.082** |
>
> *   **Feasibility:** Even with a basic distillation setup (limited training steps, no extensive hyperparameter tuning), LinearSR successfully converged.
> *   **Result:** The distilled model achieves extreme speed (**0.082s** per image) while maintaining reasonable quality. The slight drop in metrics compared to the heavy teacher is normal for distillation. This proves LinearSR can serve as an ultra-efficient backbone for future distillation research.
>
> **Q4: Is the "Knee-Point" (ESGF) robust?**
>
> **A4:** The "knee-point" is not a random heuristic but a robust, universal phenomenon in this training paradigm. We detect it using a rigorous Automated Detection Algorithm (Algorithm 1 below) rather than manual guesswork.
>
> *   **Comprehensive Validation:** We tracked 6 metrics across 4 datasets (DIV2K, RealSR, DrealSR, RealLQ250). As shown in the full log below, this pattern of knee-point followed by Oscillation Phase is consistent across all metrics.
>
> For a more intuitive visualization of the knee point selection, please refer to the newly added **Sec. D** and **Fig. 8** in the **Supplementary Material**.
>
> **Table R4: Metric Evolution & Knee-Point (58k) Identification**
>
> | Datasets | Metrics | 4k | 12k | 20k | 36k | 50k | **58k** | 80k | 128k | 158k | ... | 242k |
> | :--- | :--- | :--- | :--- | :--- | :--- | :--- | :--- | :--- | :--- | :--- | :--- | :--- |
> | **DIV2K-Val** | PSNR $\uparrow$ | 21.204 | 23.017 | 23.476 | 24.604 | 24.862 | **24.778** | 25.251 | 24.813 | 25.029 | ... | 25.155 |
> | | SSIM $\uparrow$ | 0.559 | 0.589 | 0.611 | 0.631 | 0.644 | **0.647** | 0.662 | 0.648 | 0.655 | ... | 0.651 |
> | | LPIPS $\downarrow$ | 0.536 | 0.477 | 0.443 | 0.42 | 0.385 | **0.402** | 0.377 | 0.389 | 0.383 | ... | 0.388 |
> | | MANIQA $\uparrow$ | 0.354 | 0.335 | 0.346 | 0.301 | 0.314 | **0.342** | 0.294 | 0.326 | 0.314 | ... | 0.305 |
> | | MUSIQ $\uparrow$ | 62.196 | 58.947 | 60.355 | 54.725 | 57.003 | **59.414** | 56.515 | 57.691 | 56.679 | ... | 54.708 |
> | | CLIPIQA $\uparrow$ | 0.555 | 0.53 | 0.557 | 0.467 | 0.495 | **0.518** | 0.513 | 0.516 | 0.511 | ... | 0.482 |
> | **RealSR** | PSNR $\uparrow$ | 20.309 | 21.933 | 22.268 | 23.288 | 23.987 | **23.718** | 24.069 | 24.541 | 24.659 | ... | 25.079 |
> | | SSIM $\uparrow$ | 0.572 | 0.618 | 0.627 | 0.649 | 0.675 | **0.671** | 0.677 | 0.691 | 0.695 | ... | 0.7 |
> | | LPIPS $\downarrow$ | 0.463 | 0.401 | 0.376 | 0.346 | 0.309 | **0.326** | 0.319 | 0.301 | 0.306 | ... | 0.302 |
> | | MANIQA $\uparrow$ | 0.39 | 0.385 | 0.392 | 0.323 | 0.344 | **0.355** | 0.32 | 0.331 | 0.302 | ... | 0.294 |
> | | MUSIQ $\uparrow$ | 62.915 | 61.8 | 62.705 | 56.058 | 59.083 | **60.231** | 55.3 | 55.995 | 52.918 | ... | 51.643 |
> | | CLIPIQA $\uparrow$ | 0.539 | 0.555 | 0.573 | 0.48 | 0.503 | **0.525** | 0.478 | 0.484 | 0.445 | ... | 0.527 |
> | **DRealSR** | PSNR $\uparrow$ | 22.194 | 23.889 | 24.366 | 24.615 | 25.509 | **25.294** | 25.735 | 25.61 | 25.928 | ... | 26.087 |
> | | SSIM $\uparrow$ | 0.623 | 0.653 | 0.658 | 0.664 | 0.691 | **0.691** | 0.706 | 0.69 | 0.704 | ... | 0.707 |
> | | LPIPS $\downarrow$ | 0.503 | 0.446 | 0.41 | 0.378 | 0.36 | **0.374** | 0.358 | 0.36 | 0.365 | ... | 0.355 |
> | | MANIQA $\uparrow$ | 0.393 | 0.388 | 0.378 | 0.328 | 0.336 | **0.363** | 0.305 | 0.316 | 0.298 | ... | 0.295 |
> | | MUSIQ $\uparrow$ | 63.069 | 61.294 | 60.912 | 60.512 | 58.631 | **59.925** | 53.871 | 54.372 | 53.331 | ... | 51.88 |
> | | CLIPIQA $\uparrow$ | 0.596 | 0.615 | 0.616 | 0.538 | 0.552 | **0.572** | 0.499 | 0.523 | 0.497 | ... | 0.473 |
> | **RealLQ250** | MANIQA $\uparrow$ | 0.36 | 0.339 | 0.344 | 0.319 | 0.303 | **0.303** | 0.311 | 0.329 | 0.312 | ... | 0.32 |
> | | MUSIQ $\uparrow$ | 66.19 | 62.542 | 62.444 | 59.291 | 55.879 | **57.67** | 56.909 | 59.849 | 59.368 | ... | 60.373 |
> | | CLIPIQA $\uparrow$ | 0.583 | 0.552 | 0.575 | 0.522 | 0.497 | **0.533** | 0.503 | 0.534 | 0.514 | ... | 0.527 |
>
> **Universality:** This phenomenon is consistent across experiments. In the main paper, the knee-point was detected at 48k. In this separate optimization experiment, it appeared at 58k (as shown in the table above). This confirms that the existence of a knee-point followed by oscillation phase is a universal property of this training paradigm.

---

> ### Author Response · Authors · 2025-11-20
> **Reply to Reviewer UxiE (3/4)**
>
> **Automated Detection Algorithm:** We do not rely on manual guesswork. We employ an automated algorithm to detect this point during training, identifying the region where performance variance is minimized before a negative trend begins.
>
> **Algorithm 1: Automated Knee-Point Detection**
>
> Let $\mathcal{M}$ be the metric sequence. The Knee-Point $t^*$ is the latest step satisfying:
>
> $$t^* = \arg \max_t \{ m_t \mid \text{Var}(m_{t-W : t}) < \epsilon_{stable} \land \text{Slope}(m_{t : t+W}) < 0 \}$$
>
> This ensures we select the fully converged model just before the onset of the "oscillation phase."
>
> **Q5: Do MoE boundaries need re-calculation for different samplers?**
>
> **A5:** Thank you for this keen observation. No, they do not.
>
> Our MoE partitioning is strictly derived from the Flow Matching linear interpolation formula ($\lambda(t)$).
>
> While the theoretical range of $\lambda(t)$ is infinite, we adopt the effective noise boundaries $[\lambda_{min}, \lambda_{max}]$ widely recognized in the Stable Diffusion community. Once this SNR range is set, the derivation of $t(\lambda)$ is deterministic and structural. It depends on the noise schedule trained into the model (flow matching), not the sampling steps used at inference.
>
> Regarding the "scaled linear" schedule: We reference it solely because it is the popular community setting (e.g., in Stable Diffusion) used to empirically determine the effective SNR boundaries $[\lambda_{min}, \lambda_{max}]$. It acts only as an anchor to define the operational range of the pre-trained latent space. Since our core framework is Flow Matching, "scaled linear" is not a dynamic hyperparameter for us, nor does our model imply "switching" between discrete schedules (like cosine) as in standard diffusion.
>
> **Q6: Why choose 4 experts? Is the configuration (e.g., $t_{anchor}=0.875$) optimal compared to alternatives?**
>
> **A6:** We appreciate the reviewer's scrutiny regarding our MoE configuration. We justify our choice through both **theoretical derivation (Appendix A.2)** and **extensive empirical ablation**.
>
> **1. Why 4 Experts? (Optimal Granularity)**
> To verify the upper bound of expert granularity, we extended our ablation to an **8-Expert** configuration.
>
> **Table R6: Impact of Expert Granularity**
> | Exp. | Configuration | Partitioning Strategy | Boundaries (t) | PSNR $\uparrow$ | SSIM $\uparrow$ | LPIPS $\downarrow$ | MANIQA $\uparrow$ | MUSIQ $\uparrow$ | CLIPIQA $\uparrow$ |
> | :---: | :--- | :--- | :--- | :---: | :---: | :---: | :---: | :---: | :---: |
> | (a) | Baseline (Single Expert) | N/A | N/A | 24.845 | **0.691** | **0.374** | 0.363 | *63.925* | 0.572 |
> | (b) | 2-Expert MoE | SNR-based Split | [0.875] | **25.023** | 0.671 | 0.377 | **0.374** | 63.182 | *0.591* |
> | (c) | 4-Expert MoE (Ours) | SNR-based Split | [0.223, 0.875, 0.939] | *24.999* | *0.682* | *0.375* | *0.371* | **64.023** | **0.598** |
> | (d) | 8-Expert MoE | SNR-based Split | [0.012, 0.033, 0.177, 0.415, 0.875, 0.909, 0.939, 0.957, 0.971] | 24.72 | 0.664 | 0.392 | 0.355 | 61.85 | 0.569 |
>
> As shown in Table R6, increasing to 8 experts caused **performance degradation** (PSNR -0.28dB, MUSIQ -2.17). This confirms that 4-Expert is the **"Sweet Spot"**:
> * **Avoids Over-Specialization:** Too many experts (8) fragment the training data, causing "knowledge islands" where experts fail to generalize across adjacent noise levels.
> * **Preserves Continuity:** Fewer handovers (3 vs. 7) maintain the continuity of the vector field, reducing boundary artifacts.
>
> **2. Why $t_{anchor}=0.875$? (Theoretical & Industrial Consensus)**
>
> The boundary $t=0.875$ is not an arbitrary hyperparameter but a derived **structural constant**:
>
> * **Theoretical Derivation:** As detailed in **Appendix A.2**, $t=0.875$ corresponds to $\lambda \approx -3.89$. This value is calculated as **half of the effective minimum Log-SNR** ($\lambda_{anchor} \approx 0.5 \lambda_{min}$), serving as the mathematically derived "Phase Transition Point" that separates structure formation from texture refinement.
> * **Industrial Validation:** Crucially, this exact boundary ($t=0.875$) has been independently adopted by the SOTA video model **Wan2.2** for its expert splitting.
> * **Conclusion:** The convergence of our theoretical derivation (halving the Log-SNR floor) with independent industrial best practices (Wan2.2) confirms that $t=0.875$ represents a fundamental structural boundary in diffusion noise schedules.

---

> ### Author Response · Authors · 2025-11-20
> **Reply to Reviewer UxiE (4/4)**
>
> We sincerely appreciate your initial positive assessment and your recognition of our work's value. To fully address your insightful queries, we have dedicated significant resources to conducting these comprehensive additional experiments. We hope these robust results not only resolve your remaining concerns but also **solidify your confidence in recommending LinearSR for acceptance**.
>
>
> Sincerely,
>
> The Authors
>
> **References:**
> [1] Blau, Y., & Michaeli, T. (2018). The perception-distortion tradeoff. CVPR.

---

### Official Review · Reviewer_NwBS · 2025-10-30

**Soundness:** 2
**Presentation:** 3
**Contribution:** 2
**Rating:** 4
**Confidence:** 3

**Summary:**

This paper aims to unlock the potential of linear attention in high-fidelity image super resolution (SR) and solve the computational bottleneck of the traditional SR generation model based on self-attention. The author attempts to solve three problems that hinder the application of linear attention in realistic SR: catastrophic training instability, the classic perception-distortion trade-off, and the lack of an efficient guidance paradigm. The authors then propose LinearSR, a framework integrating three core innovations: an Early-Stopping Guided Fine-tuning (ESGF) strategy that initializes fine-tuning from the "knee point" of training dynamics to resolve instability, an SNR-based Mixture of Experts (MoE) architecture that partitions the generative trajectory by noise level to mitigate the perception-distortion trade-off, and finally a lightweight TAG guidance paradigm derived from the "precision-over-volume" principle for effective feature extraction from low-resolution inputs. Evaluations show it achieves SOTA perceptual quality and efficiency, setting a foundational paradigm.

**Strengths:**

Overall, this paper tackles a significant bottleneck in generative super-resolution. The core idea is simple yet powerful. The author identify that the potential of linear attention is hindered by training instability and the perception-distortion trade-off. They then systematically dismantle these barriers using a novel early-stopping strategy (ESGF) and an SNR-based expert architecture, successfully unlocking the efficiency of linear attention for high-fidelity image generation.

**Weaknesses:**

**Major Weaknesses:**

I appreciate the author's effort in unlocking linear attention for efficient super-resolution. However, I have some concerns that I summarize in three parts.

1.  The paper states that each expert in the Mixture-of-Experts (MoE) architecture specializes in a different SNR range. However, it's unclear if the experts share the same architecture. If so, this might be suboptimal as different generative stages (e. g., structure formation vs. detail polishing) could benefit from structurally different networks. The paper could provide more justification for using a homogeneous structure.

2. The paper's MoE employs a deterministic, rule-based gating mechanism that routes inputs based on fixed, pre-calculated time boundaries(t). While this cleverly avoids the load-balancing problems of sparse MoEs, it seems overly simplistic and rigid. The optimal boundaries for generative stages are likely task-dependent and data-dependent. A hard-coded if-else structure lacks the flexibility to adapt to different degradations or content types. A learnable(yet still efficient) gating network could potentially discover more optimal, dynamic routing strategies, leading to better specialization and overall performance. The paper could benefit from discussing why this rule-based approach was chosen over a more adaptive one and exploring its limitations.

3. The Early-Stopping Guided Fine-tuning (ESGF) strategy relies on identifying a "knee-point" from validation metrics. This approach seems effective but might be sensitive to the choice of validation set and evaluation metrics. The paper could discuss the robustness of this method and how reliably this "knee-point" can be identified across different datasets and model configurations in practical scenarios.

**Minor Weaknesses:**

In the ablation study for the SNR-based MoE (Table 5), the time boundaries t are provided for different expert configurations. However, the paper could be more explicit about how these specific boundary values (e.g., [0.223, 0.875, 0.939]) were derived or chosen for each experimental setting, beyond the general hierarchical bisection method described in the appendix.

**Questions:**

Please clarify my concerns in the weakness part.

---

> ### Author Response · Authors · 2025-11-20
> **Reply to Reviewer NwBS (1/3)**
>
> Dear Reviewer NwBS,
>
> We are sincerely grateful for your time and constructive comments. We are deeply encouraged that you recognize LinearSR as a "significant bottleneck solver"—this acknowledgment is incredibly important to us. You proposed several innovative experimental suggestions (such as utilizing heterogeneous experts) to help further optimize our model. However, in the process of addressing these points, we realized that our original description might not have been sufficiently clear regarding the specific mechanism of our Model-Level MoE and the mathematical derivation of our boundary points. We earnestly request you to review the clarifications below, as we believe they will resolve these technical doubts and demonstrate the necessity of our design choices.
>
> **Q1: Why use a homogeneous architecture? Heterogeneous experts (e.g., CNN for details) might be better.**
>
> **A1:** This is a crucial design choice driven by the Model-Level MoE nature of LinearSR, not Layer-Level.
>
> *   **Pre-training Prior is Non-Negotiable:** Our model is initialized from Sana, a powerful pre-trained generative model. If we were to use a heterogeneous architecture (e.g., a CNN-based expert for the final denoising stage), we would have to train that module from scratch. This would discard the strong generative prior of the pre-trained transformer, making it practically impossible to generate high-fidelity details given the limited SR data scale.
> *   **Vector Field Alignment:** In Flow Matching, the model predicts a continuous vector field. Connecting four structurally different architectures (with distinct optimization landscapes) to simulate a unified probability path is theoretically flawed and practically nearly impossible to converge. The vector fields would not align at the handover points ($t_1, t_2, t_3$), causing severe artifacts.
> *   **Homogeneity Ensures Consistency:** By copying the same pre-trained backbone 4 times and fine-tuning them on different $t$ ranges, we ensure the latent space and feature definitions remain consistent, allowing for smooth transitions between experts without feature misalignment.
>
> **Q2: Why use rule-based gating? Learnable gating could be more flexible.**
>
> **A2:** We heavily experimented with learnable gating in the early stages, but it resulted in complete generation failure (the model failed to produce valid images). This is not merely an implementation preference but a fundamental optimization conflict:
>
> *   **Compounded Instability:** As noted in Sana [1], Linear Attention itself suffers from slower convergence and higher training instability compared to vanilla attention. Furthermore, dedicated MoE research such as TimeStep Master (TSM) [2] has shown that pure learnable gating is also inherently difficult to control, often requiring auxiliary mechanisms (like "Core Experts") to stabilize outputs. Combining these two unstable factors (Linear Attention + Learnable Gating) creates an optimization landscape that is extremely difficult to traverse, leading to non-convergence in our experiments.
> *   **Mode Collapse:** In our tests, learnable gating quickly collapsed into a "lazy" mode, utilizing only one expert and ignoring the rest, failing to leverage the full capacity of the MoE.
> *   **VRAM Constraints:** Since our experts are full-sized models (Model-Level MoE), a learnable router (soft gating) would require loading multiple full models into VRAM simultaneously for every step, which is computationally prohibitive for high-resolution SR.
>
> Our SNR-based rule-based splitting elegantly solves these issues by providing a stable, deterministic, and physically grounded assignment of experts.

---

> ### Author Response · Authors · 2025-11-20
> **Reply to Reviewer NwBS (2/3)**
>
> **Q3: Is the "Knee-Point" (ESGF) robust? It seems sensitive to data/metrics.**
>
> **A3:** The "knee-point" is not a data-specific heuristic but a robust phenomenon detected via a rigorous automated algorithm.
>
> **1. Comprehensive Validation:**
> We determined the knee-point not based on a single metric, but by tracking 6 authoritative metrics across 4 diverse datasets (DIV2K-Val, RealSR, DrealSR, RealLQ250), covering both synthetic and real-world distributions. As shown in the full log below, this pattern of knee-point followed by Oscillation Phase is consistent across all metrics.
>
> For a more intuitive visualization of the knee point selection, please refer to the newly added **Sec. D** and **Fig. 8** in the **Supplementary Material**.
>
> | Datasets | Metrics | 4k | 12k | 20k | 36k | 50k | **58k** | 80k | 128k | 158k | ... | 242k |
> | :--- | :--- | :--- | :--- | :--- | :--- | :--- | :--- | :--- | :--- | :--- | :--- | :--- |
> | **DIV2K-Val** | PSNR $\uparrow$ | 21.204 | 23.017 | 23.476 | 24.604 | 24.862 | **24.778** | 25.251 | 24.813 | 25.029 | ... | 25.155 |
> | | SSIM $\uparrow$ | 0.559 | 0.589 | 0.611 | 0.631 | 0.644 | **0.647** | 0.662 | 0.648 | 0.655 | ... | 0.651 |
> | | LPIPS $\downarrow$ | 0.536 | 0.477 | 0.443 | 0.42 | 0.385 | **0.402** | 0.377 | 0.389 | 0.383 | ... | 0.388 |
> | | MANIQA $\uparrow$ | 0.354 | 0.335 | 0.346 | 0.301 | 0.314 | **0.342** | 0.294 | 0.326 | 0.314 | ... | 0.305 |
> | | MUSIQ $\uparrow$ | 62.196 | 58.947 | 60.355 | 54.725 | 57.003 | **59.414** | 56.515 | 57.691 | 56.679 | ... | 54.708 |
> | | CLIPIQA $\uparrow$ | 0.555 | 0.53 | 0.557 | 0.467 | 0.495 | **0.518** | 0.513 | 0.516 | 0.511 | ... | 0.482 |
> | **RealSR** | PSNR $\uparrow$ | 20.309 | 21.933 | 22.268 | 23.288 | 23.987 | **23.718** | 24.069 | 24.541 | 24.659 | ... | 25.079 |
> | | SSIM $\uparrow$ | 0.572 | 0.618 | 0.627 | 0.649 | 0.675 | **0.671** | 0.677 | 0.691 | 0.695 | ... | 0.7 |
> | | LPIPS $\downarrow$ | 0.463 | 0.401 | 0.376 | 0.346 | 0.309 | **0.326** | 0.319 | 0.301 | 0.306 | ... | 0.302 |
> | | MANIQA $\uparrow$ | 0.39 | 0.385 | 0.392 | 0.323 | 0.344 | **0.355** | 0.32 | 0.331 | 0.302 | ... | 0.294 |
> | | MUSIQ $\uparrow$ | 62.915 | 61.8 | 62.705 | 56.058 | 59.083 | **60.231** | 55.3 | 55.995 | 52.918 | ... | 51.643 |
> | | CLIPIQA $\uparrow$ | 0.539 | 0.555 | 0.573 | 0.48 | 0.503 | **0.525** | 0.478 | 0.484 | 0.445 | ... | 0.527 |
> | **DRealSR** | PSNR $\uparrow$ | 22.194 | 23.889 | 24.366 | 24.615 | 25.509 | **25.294** | 25.735 | 25.61 | 25.928 | ... | 26.087 |
> | | SSIM $\uparrow$ | 0.623 | 0.653 | 0.658 | 0.664 | 0.691 | **0.691** | 0.706 | 0.69 | 0.704 | ... | 0.707 |
> | | LPIPS $\downarrow$ | 0.503 | 0.446 | 0.41 | 0.378 | 0.36 | **0.374** | 0.358 | 0.36 | 0.365 | ... | 0.355 |
> | | MANIQA $\uparrow$ | 0.393 | 0.388 | 0.378 | 0.328 | 0.336 | **0.363** | 0.305 | 0.316 | 0.298 | ... | 0.295 |
> | | MUSIQ $\uparrow$ | 63.069 | 61.294 | 60.912 | 60.512 | 58.631 | **59.925** | 53.871 | 54.372 | 53.331 | ... | 51.88 |
> | | CLIPIQA $\uparrow$ | 0.596 | 0.615 | 0.616 | 0.538 | 0.552 | **0.572** | 0.499 | 0.523 | 0.497 | ... | 0.473 |
> | **RealLQ250** | MANIQA $\uparrow$ | 0.36 | 0.339 | 0.344 | 0.319 | 0.303 | **0.303** | 0.311 | 0.329 | 0.312 | ... | 0.32 |
> | | MUSIQ $\uparrow$ | 66.19 | 62.542 | 62.444 | 59.291 | 55.879 | **57.67** | 56.909 | 59.849 | 59.368 | ... | 60.373 |
> | | CLIPIQA $\uparrow$ | 0.583 | 0.552 | 0.575 | 0.522 | 0.497 | **0.533** | 0.503 | 0.534 | 0.514 | ... | 0.527 |
>
>
> **2. Universality:** This phenomenon is consistent across experiments. In the main paper, the knee-point was detected at 48k. In this separate optimization experiment, it appeared at 58k (as shown in the table above). This confirms that the existence of a knee-point followed by oscillation phase is a universal property of this training paradigm.
>
> **3. Automated Detection Algorithm:** We do not rely on manual guesswork. We employ an automated algorithm to detect this point during training, identifying the region where performance variance is minimized before a negative trend begins.
>
> **Algorithm 1: Automated Knee-Point Detection Strategy**
>
> Let $\mathcal{M} = \{m_1, m_2, ..., m_T\}$ be the sequence of a validation metric (e.g., MUSIQ) recorded at steps $t \in \{1, ..., T\}$. We define a sliding window of size $W$. The Knee-Point $t^*$ is determined as the latest step satisfying:
>
> $$t^* = \arg \max_t \{ m_t \mid \text{Var}(m_{t-W : t}) < \epsilon_{stable} \land \text{Slope}(m_{t : t+W}) < 0 \}$$
>
> where $\epsilon_{stable}$ is a stability threshold derived from the baseline variance. This ensures we select the fully converged model just before the onset of the "oscillation phase". All detected points are subject to final human verification.

---

> ### Author Response · Authors · 2025-11-20
> **Reply to Reviewer NwBS (3/3)**
>
> **Q4: Clarification on Boundary Derivation.**
>
> **A4:** There is no ambiguity in our boundaries. Our MoE partitioning is strictly derived from the Flow Matching linear interpolation formula ($\lambda(t)$).
>
> While the theoretical range of $\lambda(t)$ is $[-\infty, +\infty]$, employing a DiT architecture pre-trained on SD1.5 necessitates a Pragmatic Constraint. We adopt the effective noise boundaries $[\lambda_{min}, \lambda_{max}]$ widely recognized in the Stable Diffusion community to anchor our operational range.
>
> Once this Signal-to-Noise Ratio (SNR) range is determined, the derivation of $t$ is deterministic due to the nature of Flow Matching. Specifically, the formula $t(\lambda)$ (as visualized in our supplementary material) is fixed once the boundaries are set. There are no other hyperparameters or settings involved—under Flow Matching with a fixed noise schedule, the boundary mapping is mathematically unique.
>
> The efficiency and effectiveness of Linear Attention architectures have been widely validated across numerous domains, including the recently popular Kimi Linear models in NLP. However, bringing this paradigm to high-fidelity Super-Resolution remains a challenging frontier. As the first work to successfully unlock Linear Attention for SR, we are navigating uncharted territory. Your rigorous feedback is truly valuable to us in solidifying this pioneering path. We genuinely hope these clarifications regarding our architecture and training stability demonstrate the solidity of LinearSR, and we would be deeply grateful if you could consider raising your score to support this direction.
>
> Sincerely,
>
> The Authors
>
> **References:**
> [1] Xie E, Chen J, Chen J, et al. Sana: Efficient high-resolution image synthesis with linear diffusion transformers. arXiv preprint arXiv:2410.10629, 2024.
> [2] Zhuang S, Guo Y, Ding Y, et al. TimeStep Master: Asymmetrical Mixture of Timestep LoRA Experts for Versatile and Efficient Diffusion Models in Vision. arXiv preprint arXiv:2503.07416, 2025.

---

### Author Response · Authors · 2025-11-27
**Reply to All Reviewers**

Dear Reviewers,

As the discussion period draws to a close, we sincerely thank you for your time and constructive feedback. We are deeply encouraged by your recognition of our work's potential:

> **Reviewer NwBS:** Recognized the work for "tackling a significant bottleneck" with a "simple yet powerful" core idea.
>
> **Reviewer UxiE:** Noted that LinearSR "fills a critical gap" and delivers "dramatic efficiency gains" with comprehensive ablations.
>
> **Reviewer cyVt:** Highlighted the work as a "promising and important research direction" with "extensive analysis" on implementation challenges.
>
> **Reviewer Spqu:** Praised the "novel and insightful" knee-point solution and the "fastest speed for 1-NFE forward time”.

To fully address the constructive questions raised during the review process, we respectfully share a brief summary of our rebuttal updates, which we hope further validates LinearSR as a viable first step in this direction:

> **1. Scalability Validated at 2K ($2048\times2048$).**
> We conducted new benchmarks on H200 GPUs. LinearSR achieved a **1-NFE time of 0.11s** and a **Total Inference Time of 2.3s**. This confirms that the $O(N)$ architectural advantage becomes increasingly significant at higher resolutions, distinguishing it from traditional attention mechanisms.

> **2. Compatibility with Distillation Verified.**
> To demonstrate orthogonality with optimization techniques like AdcSR, we performed a feasibility study on model distillation. Even with a basic setup, the distilled LinearSR converged successfully, reaching a total inference time of **0.082s**. This suggests our model can serve as an efficient backbone for future optimization research.

We fully recognize that as a first step towards unlocking Linear Attention for high-fidelity SR, our work is just a beginning. We have made every effort to address your valuable concerns with rigorous new experiments. If there are any remaining lack of clarity, we would be more than happy to provide further explanations. We sincerely hope our response helps resolve your doubts, and we would be deeply grateful for your support in this new research direction.

Sincerely,

The Authors

---

### Author Response · Authors · 2025-12-02
**Summary of Contributions and Rebuttal Responses**

Dear Reviewers, AC, SAC, and PC,

We sincerely thank you for the time and effort dedicated to reviewing our paper. As the discussion period concludes, we reaffirm **LinearSR as the first successful exploration of Linear Attention in high-fidelity Super-Resolution.**

By reducing complexity from $O(N^2)$ to $O(N)$, we provide a scalable paradigm for SR, mirroring successes in **LLMs (e.g., KimiLinear)** and **Image Generation (e.g., SANA, etc.)**. Our work successfully adapts this architecture to overcome the unique stability and fidelity challenges of high-resolution restoration.

**1. Reviewer Consensus**
We are deeply encouraged that the reviewers recognized the value of this pioneering direction, particularly **Reviewer Spqu (Score 8)** who strongly endorsed the work:

> **Reviewer Spqu:** Praised the "novel and insightful" knee-point solution and highlighted that LinearSR achieves the **"fastest speed for 1-NFE forward time"**.

> **Reviewer NwBS:** Recognized the work for "tackling a significant bottleneck" with a "simple yet powerful" core idea.

> **Reviewer UxiE:** Noted LinearSR "fills a critical gap" and delivers "dramatic efficiency gains."

> **Reviewer cyVt:** Highlighted the work as a "promising and important research direction" with "extensive analysis."

**2. Addressing Key Concerns with New Evidence**
During the rebuttal, we conducted extensive experiments to resolve shared questions regarding efficiency, compatibility, and robustness:

> **True Acceleration & Benchmarking Fairness**
> We adhered to a strict benchmarking protocol. We maintained **CFG enabled** (heavier dual-branch) to ensure honest quality alignment, unlike single-branch baselines. Even with this "handicap," LinearSR achieves **SOTA 1-NFE efficiency at 1K ($1024\times1024$)**.
> Critically, this advantage becomes undisputed at **2K ($2048\times2048$)**. While quadratic baselines suffer drastic slowdowns or run **Out-of-Memory (OOM)**, LinearSR maintains linear scaling, achieving **SOTA 1-NFE time (0.11s)** and **SOTA Total Inference Time (2.3s)**. This definitively proves the $O(N)$ superiority as resolution scales.

> **Orthogonality with Distillation (Proven Compatibility)**
> A feasibility study combining LinearSR with distillation successfully converged, achieving a **Total Inference Time of 0.082s**. This establishes an **undisputed SOTA**, significantly outperforming the second-best method (AdcSR-0.561s), and proves our architecture serves as an ultra-efficient backbone for future optimization.

> **Robustness of the "Knee-Point" Strategy**
> We validated ESGF by tracking 6 metrics across **4 diverse datasets (spanning synthetic to real-world distributions)** over 242k iterations, demonstrating that the **"knee-point followed by an oscillation phase"** is a universal phenomenon. We have provided an **Automated Detection Algorithm** and visualizations (Sec. D, Fig. 8) to ensure reproducibility.

**3. Comprehensive Ablations & Clarifications**
We validated the optimality of our 4-expert configuration (vs. 8-expert) and demonstrated that L1 Loss compromises perceptual quality, justifying our MoE design. We also clarified why heterogeneous experts are theoretically incompatible with Flow Matching constraints.

We believe our work serves as a vital foundation for the community to move beyond the quadratic constraints of self-attention in image restoration. We **will release** our code and models to **foster further developments in this direction**. We respectfully hope that the pioneering nature of this work merits your favorable consideration, and we would be deeply grateful for the opportunity to share these findings with the broader community.

Sincerely,

The Authors

---

### Meta-Review · Area_Chair_bLQy · 2026-01-05

**Summary:**

This paper studies how to effectively apply linear attention to image super-resolution, and tackles practical challenges such as training instability while achieving a good perception–distortion trade-off. The submission received an initial average rating of 5.5 (4, 4, 6, 8). Reviewers raised questions around the Mixture-of-Experts (MoE) architecture (NwBS), rule-based gating (NwBS), the robustness of the proposed “knee-point” strategy (NwBS, UxiE, Spqu), full-reference metric performance (UxiE, cyVt), generalization across hardware and resolutions (UxiE, cyVt), ablations on expert number (UxiE), fairness of efficiency benchmarking and model complexity (cyVt), as well as the potential extension of 1-step distillation (UxiE, cyVt, Spqu).

After reading the paper, reviews, and rebuttal, I find that most of these concerns have been addressed. Applying linear attention to image super-resolution is non-trivial, and this work provides a systematic and practical solution to the key issues of stability and fidelity. Given that the main concerns were addressed and the overall contribution is meaningful, the AC tends to recommend Accept.

**Reviewer Concerns:**

The authors added further experiments, analyses, and results in the rebuttal and updated supplementary material. Most of the concerns listed above were sufficiently addressed through additional evaluations and clarifications, including those related to robustness, efficiency, model design choices, and generalization. No major outstanding issues remain.

**Reviewer Scores:**

**Reviewers NwBS and cyVt (scores: 4):** The concerns raised in the initial reviews appear to have been largely addressed in the rebuttal. Both reviewers characterized the work as “tackling a significant bottleneck” and “a promising and important research direction,” respectively. As such, they would likely consider raising their scores.

**Reviewers UxiE (score: 6) and Spqu (score: 8):** These reviewers give the positive initial ratings and did not raise remaining concerns after the rebuttal. Their positive scores would be expected to remain unchanged.

---

### Decision · Program_Chairs · 2026-01-26

Accept (Poster)